# Photonic comb-rooted synthesis of ultra-stable terahertz frequencies

Dong-Chel Shin [1], Byung Soo Kim[1], Heesuk Jang[1], Young-Jin Kim [1] ✉ & Seung-Woo Kim [1] ✉

Stable terahertz sources are required to advance high-precision terahertz applications such as molecular spectroscopy, terahertz radars, and wireless communications. Here, we demonstrate a photonic scheme of terahertz synthesis devised to bring the well-established feat of optical frequency comb stabilization down to the terahertz region. The source comb is stabilized to an ultra-low expansion optical cavity offering a frequency instability of $10^{-15}$ at 1-s integration. By photomixing a pair of comb lines extracted coherently from the source comb, terahertz frequencies of 0.10–1.10 THz are generated with an extremely low level of phase noise of −70 dBc/Hz at 1-Hz offset. The frequency instability measured for 0.66 THz is $4.4 \times 10^{-15}$ at 1-s integration, which reduces to $5.1 \times 10^{-17}$ at 65-s integration. Such unprecedented performance is expected to drastically improve the signal-to-noise ratio of terahertz radars, the resolving power of terahertz molecular spectroscopy, and the transmission capacity of wireless communications.

Terahertz wave sources are needed in diverse applications such as molecular spectroscopy[1–3], terahertz radar[4], and wireless communications[5–7], with increasing demands on frequency stability and accuracy. Particularly, for next-generation wireless communications, high-precision sources are required to certify relevant terahertz equipment and devices with traceability to the atomic RF or optical clocks[6,8,9]. Such accurate terahertz sources are also required to enable high-density data transmission by featuring the wavelength-polarization multiplexing with higher-order signals modulation, e.g., 32-QAM (quadrature amplitude modulation)[5,10,11]. As for terahertz radars, low-noise terahertz sources are essential to provide higher signal-to-noise detection sensitivity with less vulnerability to weather and ambient light than LIDAR (light detection and ranging)[12,13]. In addition, for terahertz spectroscopy to be able to detect suppressed molecular rotational and vibrational modes[14,15] or extraordinary behaviors of metamaterials[16], terahertz sources narrowed to a sub-kHz linewidth are considered indispensable[17].

Stable, accurate terahertz sources have evolved over the past two decades along with the emergence of the optical frequency comb as a photonic frequency divider[18–20]. An outstanding example is terahertz quantum cascade lasers (QCLs) which can be either phase-locked directly to a frequency comb[21,22] or injection-locked to a stable cw terahertz radiation synthesized from a frequency comb[23,24]. Meanwhile, direct optical heterodyning of cw lasers through a photomixer, such as unitraveling-carrier photodiodes (UTC-PDs)[25,26] or high-speed p-i-n photodiodes[27–29], enables effective production of stable terahertz waves[9,30–35]. Through such photomixers, soliton microcombs stabilized with an exceptionally high repetition rate can also be used for the generation of terahertz waves with suppressed phase noise by optical frequency division[36–38]. The best level of phase noise and frequency stability achieved yet reached −40 dBc/Hz at 1 Hz and $2 \times 10^{-13}$ at 1 s, respectively, which were obtained by relying on the microwave H-maser[36].

In this study, with the aim to enhance the frequency stability of terahertz generation up to the state-of-the-art precision of optical clocks, we devise a photonic synthesizer that permits coherent optical-to-terahertz down-conversion directly from a source optical comb. Our synthesizer utilize an erbium-doped fiber oscillator as the source comb with locking to the resonance peak of an ultra-low expansion optical cavity. With further self-referencing via $f–2f$ interferometry, the source comb is configured to provide its comb lines, by injection locking to laser diodes, with a fractional frequency instability of $10^{-15}$ at

[1]Department of Mechanical Engineering, Korea Advanced Institute of Science and Technology (KAIST), 291 Daehak-ro, Yuseong-gu, Daejeon 34141, Republic of Korea. ✉e-mail: yj.kim@kaist.ac.kr; swk@kaist.ac.kr

1-s integration over a 4 THz tunable range. With active removal of the random thermal noise occurring along the fiber delivery line, terahertz waves are generated by heterodyne photomixing of a pair of comb lines with the phase noise and frequency stability below −70 dBc/Hz at 1-Hz offset and $4.4 \times 10^{-15}$ at 1-s integration, respectively. With such exceptional performance, our terahertz synthesizer is intended to act as a terahertz clock facilitating unprecedented terahertz applications such as extremely-low noise radars, next-generation 6G wireless communications, and high-resolution scientific molecular spectroscopy.

## Results

### Hardware system configuration for optical-to-terahertz down-conversion

Figure 1a illustrates the concept of the terahertz synthesizer devised in this investigation. The central element is an optical frequency comb constructed by an Er-doped fiber oscillator to provide optical frequencies for down-conversion to terahertz frequencies by heterodyne photomixing. The source frequency comb is adjusted to yield an initial repetition rate near 100 MHz and stabilized to a high-finesse optical cavity carefully prepared with ultra-low expansion (ULE) glass to deli-

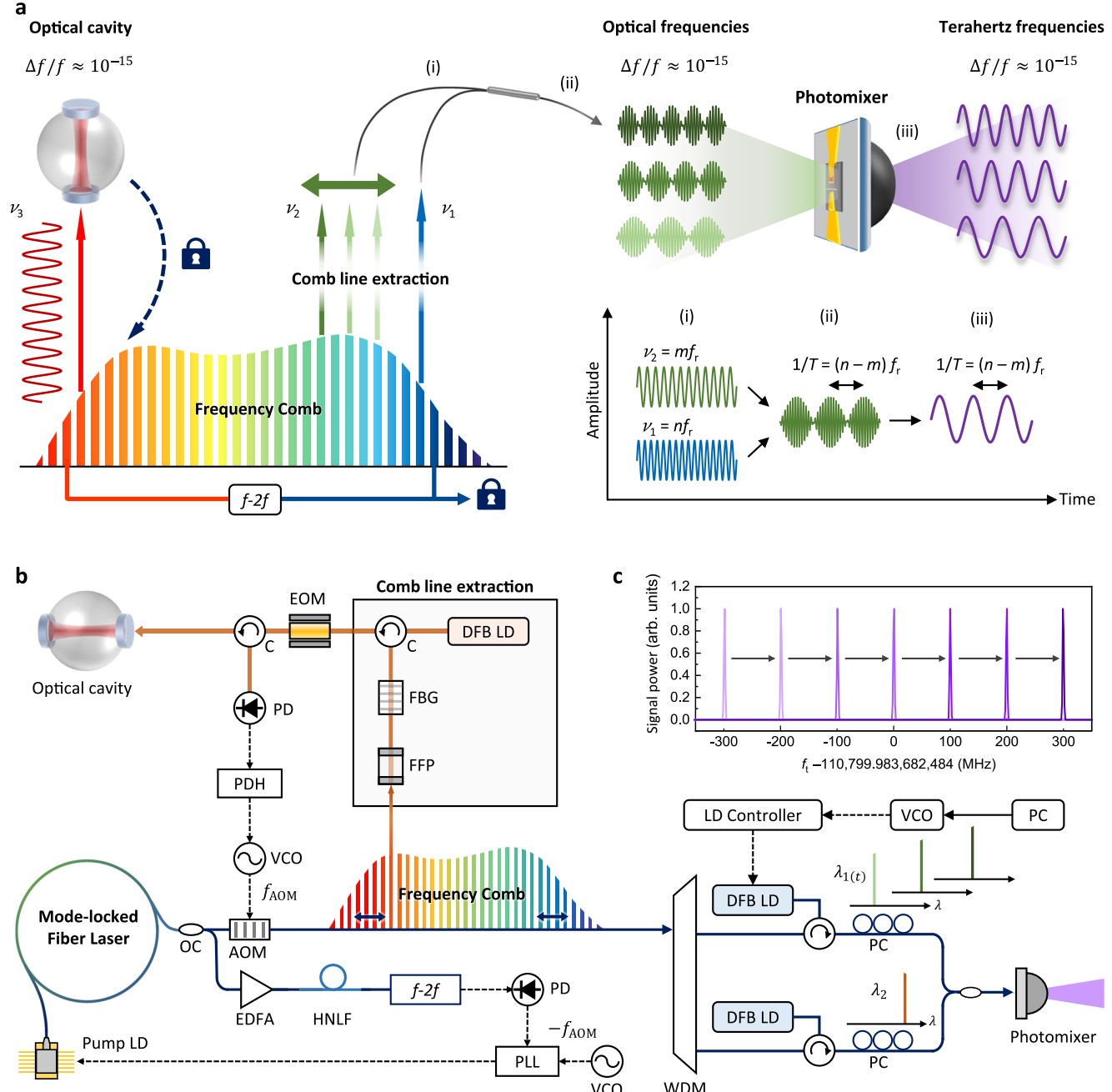

**Fig. 1 | Comb-based generation of terahertz frequencies with reference to an optical cavity. a** Multiple pairs of optical frequencies are extracted from an optical frequency comb (OFC) by means of injection locking to laser diodes (LDs). The extracted optical frequency pairs are converted to terahertz frequencies through a photomixer, while the OFC is locked to a high-finesse cavity. $f_r$: repetition rate, T: period. **b** Detailed scheme of comb-based terahertz synthesizer. **c** Terahertz synthesis with 100 MHz steps. AOM acousto-optic modulator, DFB distributed feedback, EDFA erbium-doped fiber amplifier, EOM electro-optic modulator, FFP fiber Fabry-Perot filter, HNLF highly nonlinear fiber, LD laser diode, OC optical coupler, PC polarization controller, PD photodetector, PDH Pound-Drever-Hall control, PLL phase-locked loop, VCO voltage-controlled oscillator.

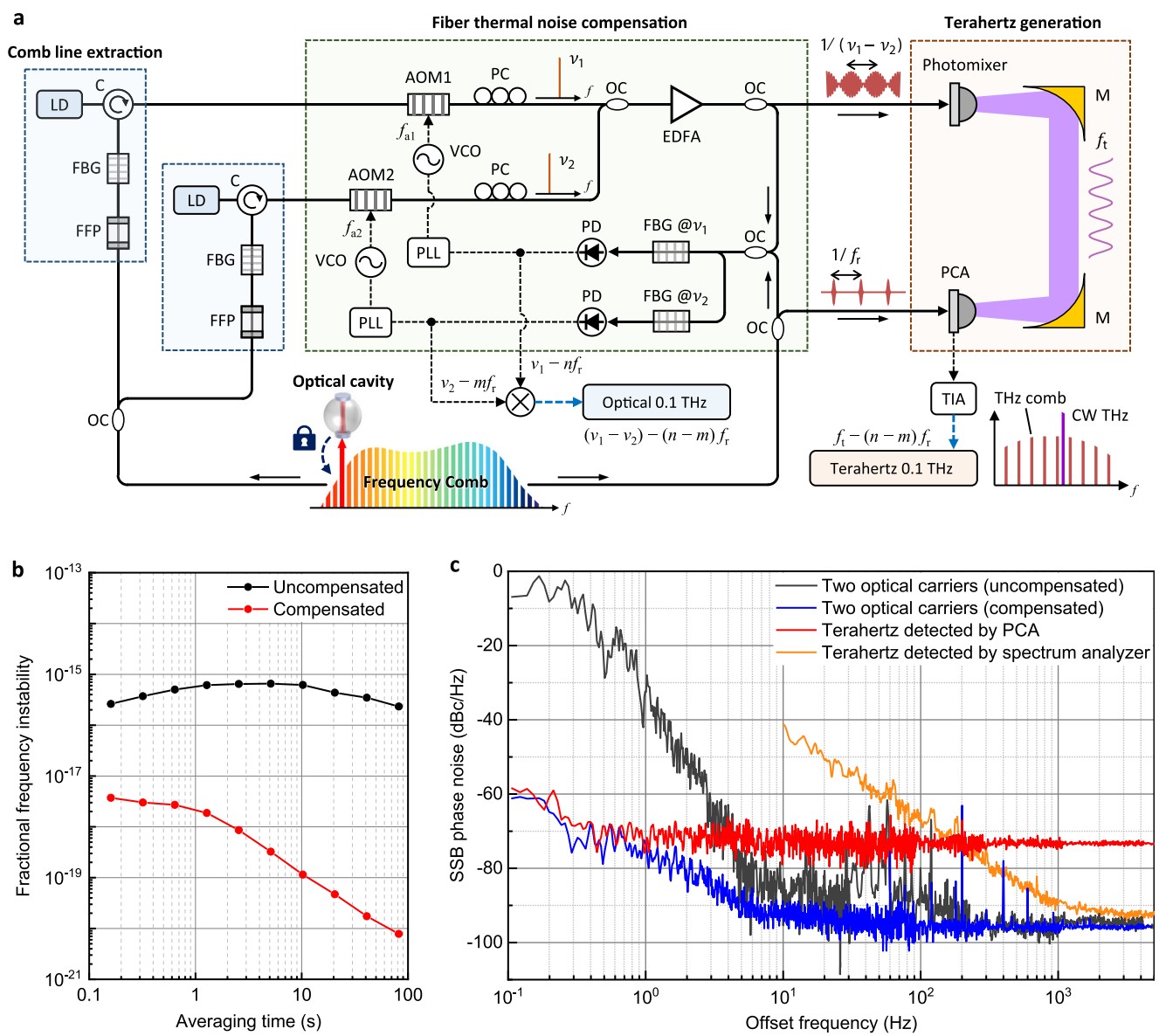

**Fig. 2 | Phase noise suppression for terahertz synthesis. a** Fiber thermal noise compensation. **b** Allan deviations of comb lines with and without thermal noise compensation. **c** Phase noise spectra. AOM acousto-optic modulator, C optical circulator, EDFA erbium-doped fiber amplifier, EOM electro-optic modulator, FBG fiber Bragg grating filter, FFP fiber Fabry-Perot filter, LD laser diode, M off-axis parabolic mirror, OC optical coupler, PC polarization controller, PCA photo-conductive antenna, PD photodetector, PDH Pound-Drever-Hall technique, PLL phase-locked loop, TIA transimpedance amplifier, VCO voltage-controlled oscillator, WDM wavelength division multiplexer.

ver a frequency stability of $\Delta f/f \approx 10^{-15}$ at 1-s averaging[39]. The high-finesse optical cavity provides resonant transmission lines of 8 kHz bandwidth (FWHM), corresponding to a finesse value of $4 \times 10^5$ or a quality factor of $2 \times 10^{10}$ (see "Methods" for more details). Note that, by incorporating one of the state-of-the-art optical clocks[40], the frequency stability may be extended to a long-term level of $\Delta f/f \approx 10^{-17}$ or even better, at 1000 s or longer integration[41], although not implemented here in the current version of the hardware system. Secondly, from the stabilized source comb, a pair of comb lines are selected so that their frequency difference is converted to a terahertz output, in the range of 0.1 to 1.0 THz, by means of heterodyne mixing through a photomixer based on an InGaAs p-i-n photodiode.

## Source comb stabilization
Figure 1b shows in detail how the source comb is configured to deliver stable comb lines for optical-to-terahertz down-conversion. Specifically, multiple comb lines are selected concurrently, with each being isolated individually from its neighbors through an optically narrow bandpass filter devised by combining a fiber Fabri-Perot (FFP) etalon with a fiber Bragg grating (FBG)[42]. The filtered comb lines are extracted, independently one by one, by injection locking to a distributed-feedback laser diode (DFB LD) permitting power amplification to ~20 mW. It is important to note that the process of power amplification by means of injection locking causes no degradation of the original stability inherited from the source comb[39,43,44]. Now, as the initial step of comb stabilization, the whole structure of the source comb is tuned along the frequency domain through an acousto-optic modulator (AOM), so that one of the extracted comb lines is locked precisely to a resonance peak of the ULE cavity by implementing the Pound-Drever-Hall (PDH) technique[45]. The next step of comb stabilization is the detection of the carrier-envelope offset of the source comb using an f-2f interferometer with subsequent nullification control to a zero offset[39,46]. This two-step stabilization scheme (Methods for more details) leads the optical frequency $\nu_n$ of the nth comb line to be an

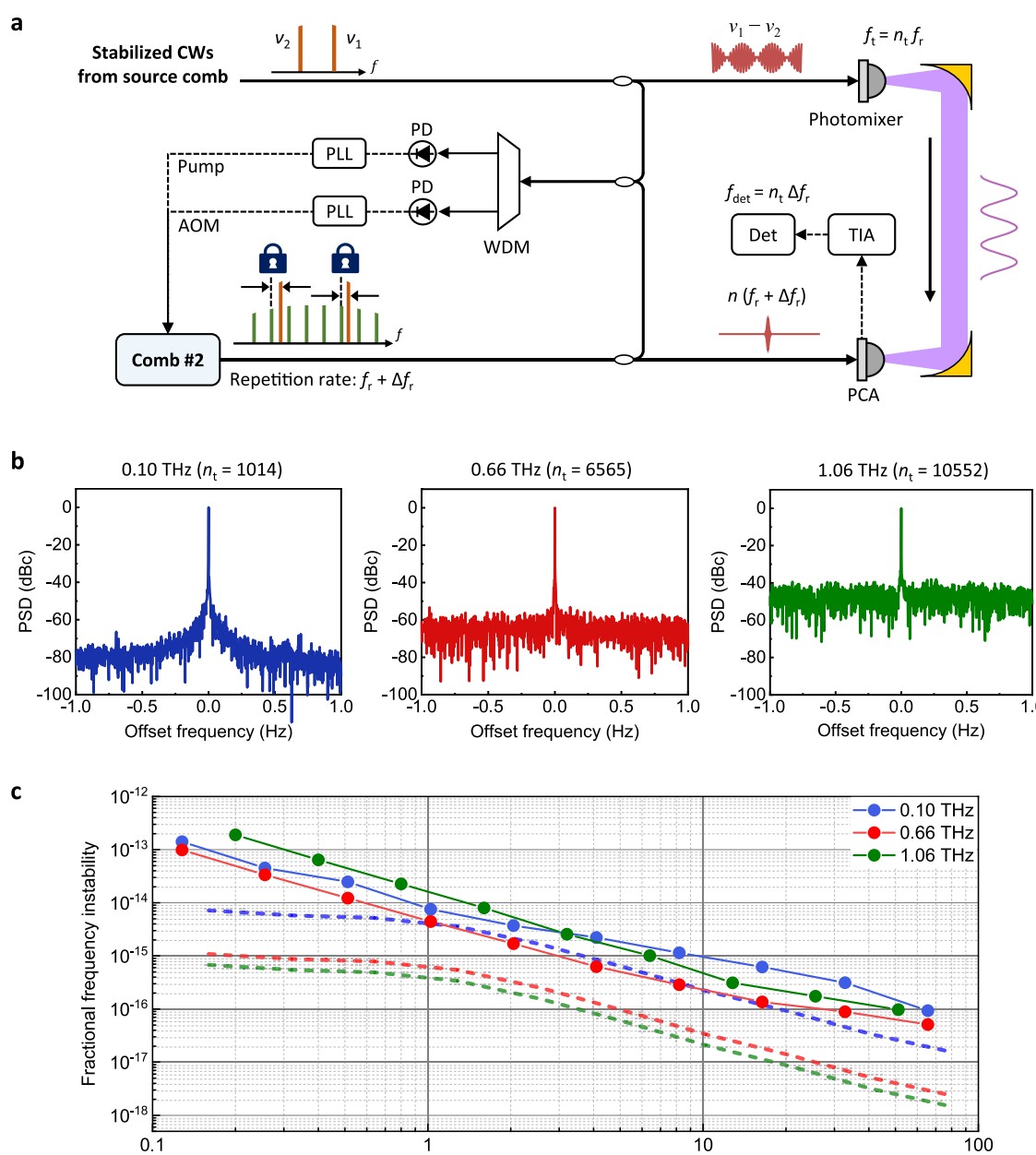

**Fig. 3 | Absolute determination of terahertz frequencies. a** Measurement scheme employing an extra comb—Comb #2. **b** Power spectral densities (PSDs) of generated terahertz frequencies. **c** Allan deviations of generated terahertz-frequencies. Dashed lines are the ideal values estimated purely from the optical phase noise of comb line pairs. AOM acousto-optic modulator, Det detection, PCA photoconductive antenna, PD photodetector, PLL phase-locked loop, TIA transimpedance amplifier, WDM wavelength division multiplexing.

integer multiple of the repetition rate $f_r$. Consequently, the terahertz frequency $f_t$ obtained by heterodyne photomixing of two different optical frequencies of $v_n = n \times f_r$ and $v_m = m \times f_r$ is defined $f_t = (n - m) \times f_r$.

**Thermal phase noise of fiber delivery line**

Figure 2a illustrates how two optical comb lines, denoted by $v_1$ and $v_2$, are delivered to generate a terahertz wave by heterodyne photomixing. Three pairs of $v_1$ and $v_2$ at 1563 and 1564 nm, 1550 and 1555 nm, and 1555 and 1563 nm are selected to produce 0.10, 0.66, and 1.05 THz, respectively (details in "Methods"). It is important to note that the frequency stability of the terahertz waves generated by photomixing is found affected by not only the source comb stability but also two extra causes; one is the thermal noise of the fiber delivery line and the other the conversion noise of the photomixer. The fiber delivery line per each comb line requires a minimum length

of 5 m in order to accommodate all the functional components—an FFP filter, an FBG filter, an injection-locking LD and an AOM—from the source comb to the photomixer. This lengthy fiber line creates the thermal phase noise of random and uncorrelated nature, causing a fractional frequency instability ($\Delta f/f$) of ~$10^{-16}$ in terms of the Allan deviation at 1–100 s integration (black curve, Fig. 2b), when compared with the original comb lines of the source comb by optical beating at the inlet location of the photomixer (Methods). This additional instability level may be considered not substantial in the optical frequency domain, but its effect in the terahertz domain becomes augmented; even though the thermal noise ($\Delta f$) remains the same, the denominator frequency ($f$) undergoes a three-order-of-magnitude down-conversion. Thus, the fiber thermal noise of each comb delivery line needs to be compensated actively through phase-locked loop (PLL) control with respect to the original

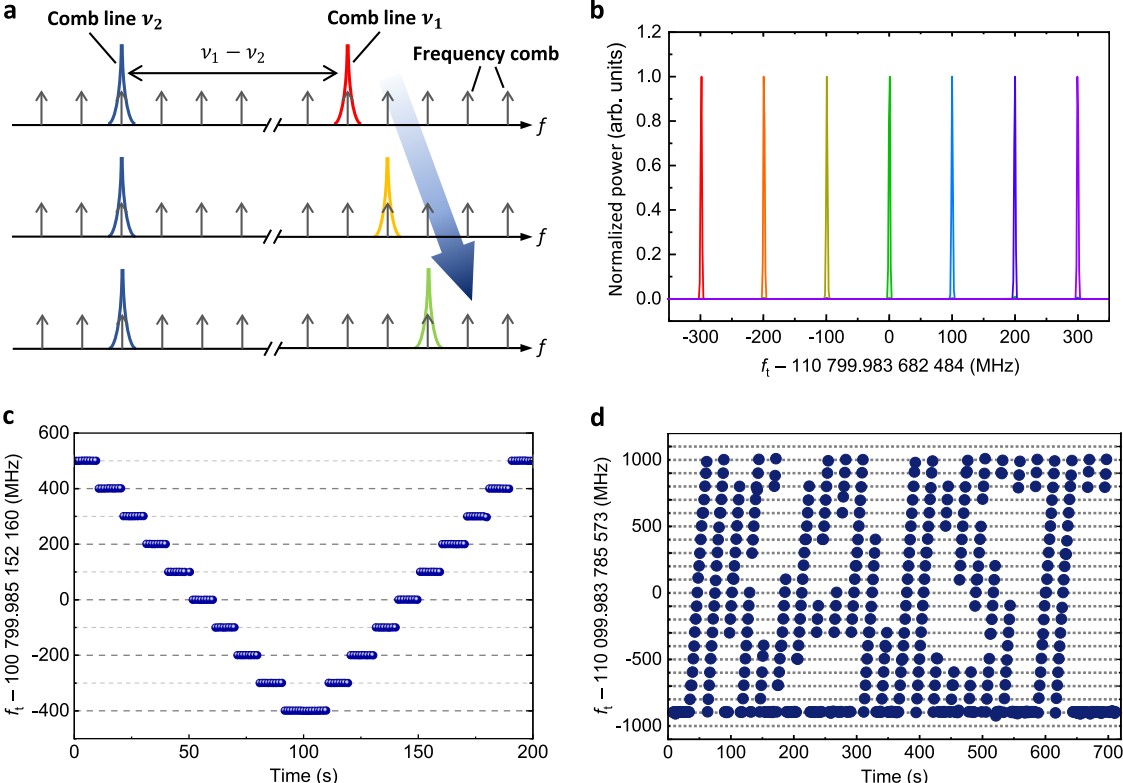

**Fig. 4 | Arbitrary terahertz frequency synthesis. a** Selective comb line extraction by injection-locking position control. **b** Normalized terahertz frequency spectra. As a demonstration, terahertz frequencies are tuned with a wide dynamic range -2 GHz over time, drawing **c** the letter V and **d** KAIST logo.

source comb (Methods). As a result, in the case of a 0.1 THz wave, the fiber thermal noise is suppressed to a fractional level of $1.87 \times 10^{-18}$ at 1 s and $7.85 \times 10^{-21}$ at 100 s integration (red curve, Fig. 2b), i.e., at least two orders of reduction. Next, the phase noise of the difference between two comb lines, i.e., $|\nu_1 - \nu_2|$, is measured in the optical regime at the inlet of the photomixer ("Methods"). This optically carried terahertz signal yields a phase noise level of as large as −32.5 dBc/Hz at 1-Hz offset when the PLL control of fiber thermal noise is not activated. In contrast, with the PLL control turned on, the phase noise reduces to −73.5 dBc/Hz at 1 Hz and further −95.4 dBc/Hz at 10 Hz. This result indicates a 30 dB improvement achieved by the PLL-controlled suppression of the fiber thermal noise (blue curve, Fig. 2c). The fiber thermal noise would be further reduced by minimizing the split length of the fiber lines needed for comb line extractions together with more rigorous regulation of the ambient temperature drift.

### Terahertz detection noise measurement

Now the phase noise of $|\nu_1 - \nu_2|$ is measured in the terahertz regime after photomixing, in which the terahertz waves are detected using a photoconductive antenna (PCA). For this, the pulse train of the source comb is partly made incident to the PCA as a gating comb to create a terahertz frequency comb[9]. As a terahertz frequency ruler, the down-converted source comb is made to interfere with the terahertz waves emitted from the photomixer (Fig. 2a), of which the lowest RF beat obtained from the PCA is monitored through a transimpedance amplifier (TIA). As revealed in its phase noise spectrum (Fig. 2c, red curve), the terahertz waves of 0.1 THz yield a phase noise level of −73 dBc/Hz as compared to that of the optically carried counterpart (Fig. 2c blue line). The elevated noise floor of the terahertz wave is attributable to the low dark resistance of the PCA[47], which shows a typical pattern of white noise remaining almost constant over offset frequencies.

### Absolute determination of terahertz frequencies

Figure 3a illustrates a hardware addition made to the PCA-based ter-ahertz detection part so as to determine the absolute frequency of the generated terahertz wave instantly. Note that the terahertz frequency shown previously in Fig. 2a was determined by subtraction of $\nu_1$ and $\nu_2$, individually measured by an optical wavelength meter. For absolute determination, the input reference comb to the PCA is changed from the source comb to another separate comb, referred to as Comb #2 for distinction. With respect to the comb lines $\nu_1$ and $\nu_2$ selected for heterodyne photomixing, Comb #2 is stabilized by incorporating two phase-locked loops (PLLs) so as to yield a repetition rate slightly different by an amount of $\Delta f_r$ from that of the source comb (details in "Methods"). The absolute mode order, $n_t = n_1 - n_2$, of the terahertz frequency generated by heterodyne mixing of $\nu_1$ and $\nu_2$ is decided as $n_t = f_{det}/\Delta f_r$ with $f_{det}$ being the RF beat monitored from the PCA as $f_{det} = n_t \Delta f_r$. Then, the exact value of the terahertz frequency $f_t$ is calculated as $f_t = n_t f_r$ with $f_r$ being the original repetition rate of the source comb.

### Terahertz stability evaluation

Three different terahertz frequencies are chosen; 0.1, 0.66 and 1.06 THz in ascending order within the operating range of the pho-tomixer. As illustrated in Fig. 3b, the generated terahertz frequencies are quantified by Fourier transform (FT) of the RF beat ($f_{det}$) obtained from the PCA (Fig. 3a). The result shows that the spectral linewidth for all the terahertz frequencies appears to be less than 2 mHz, being limited by the FT resolution in our experiment calculated with a sampling time of 500 s. It is worthwhile to note that, without the removal of the thermal noise of the fiber delivery line explained earlier (Fig. 2b), the spectral linewidth tends to broaden to 1.46 Hz, almost two orders of magnitude larger (Supplementary Information). The signal-to-noise ratio (SNR) is evaluated to be 75, 60, and 42 dB, respectively, which degrades for higher terahertz

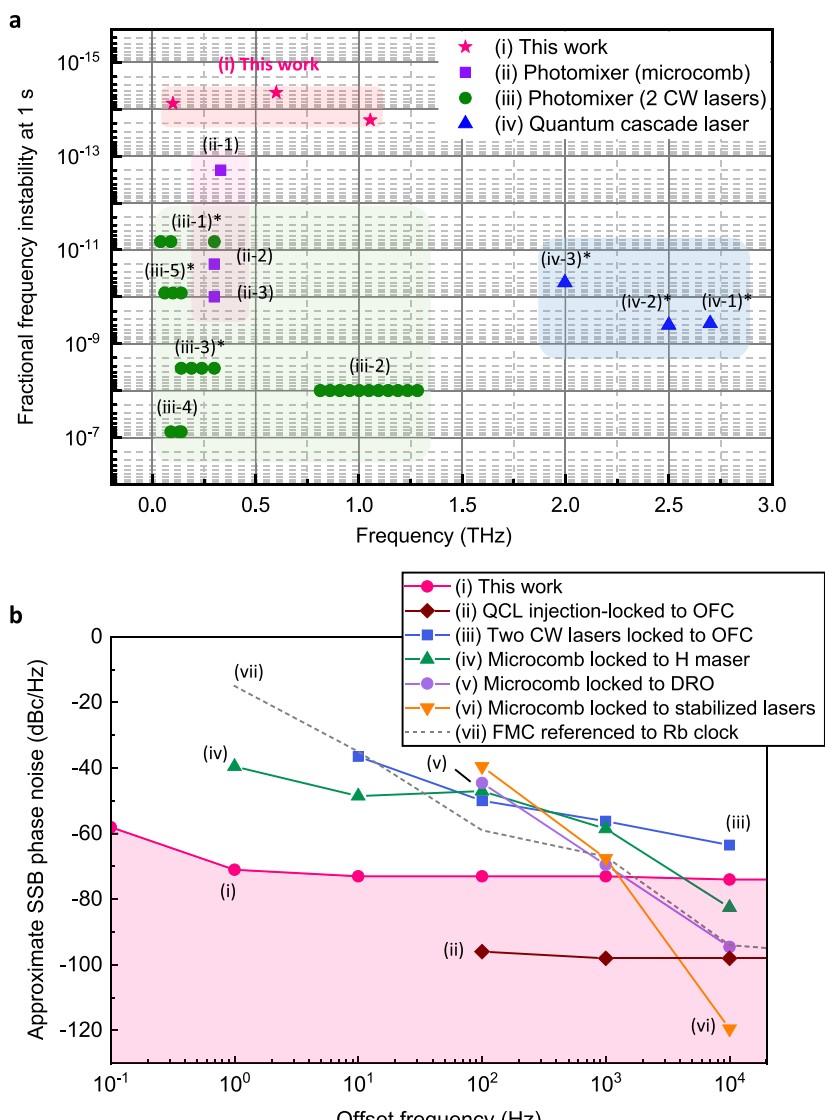

**Fig. 5 | Performance comparison with state-of-the-art terahertz sources.**
**a** Frequency instability at 1 s vs. operating frequency range; (i) this work, (ii-*n*) microcomb-based photomixing (*n* = 1–3 refer to refs. [36–38]), (iii-*n*) heterodyne mixing of cw lasers (*n* = 1–5 refer to refs. [31–35]), and (iv-*n*) QCLs (*n* = 1–3 refer to refs. [21–23]). The fractional frequency instabilities marked with an asterisk are extracted from linewidth data (see "Methods"). **b** Approximate single-sideband (SSB) phase noise; (i) this work, (ii) QCL injection-locked to a frequency comb (referenced to a Rb clock)[23], (iii) heterodyne mixing of two CW lasers phase-locked to a frequency comb (referenced to a quartz crystal oscillator)[31], (iv) 0.33 THz soliton microcomb stabilized to a hydrogen maser clock[36], (v) 0.30 THz soliton microcomb stabilized to a 10 GHz dielectric resonant oscillator (DRO)[37], (vi) 0.30 THz soliton microcomb stabilized to two optical Brillouin Stokes waves[38], and (vii) 0.1 THz signal of frequency multiplier chain (FMC) with up-conversion of an RF oscillator. For comparison, all carrier frequencies were scaled to 0.1 THz.

frequencies following the photomixer's optical-to-terahertz conversion efficiency[29]. Next, the fractional frequency instability is analyzed by monitoring the terahertz RF beat ($f_{\text{det}}$) in terms of the modified Allan deviation (Fig. 3c). The measured terahertz instabilities turn out larger than the ideal values (dashed, Fig. 3c) estimated in the optical domain from the residual frequency instability of the comb lines $\nu_1$ and $\nu_2$ after compensation of the fiber thermal nose (Fig. 2b). Higher generated terahertz frequencies suffer worse stabilities, being attributable to the phase noise contributed by not only the photomixer but also the PCA used for terahertz detection ("Methods"). In general, the measured instability lies at 10 dB for 0.1 THz, increasing to 20 dB for 1.06 THz. Nonetheless, it is important to note that the average instability reaches $4.4 \times 10^{-15}$ at 1-s integration, extending to $5.1 \times 10^{-17}$ at 65-s integration (0.66 THz, Fig. 3c), demonstrating the potential of transferring the state-of-the-art optical clock stability[48] down to the terahertz region.

## Successive tuning of terahertz frequencies

Figure 4 illustrates how our synthesizer is able to generate terahertz frequencies in a sequential scanning mode. Yet most of the existing comb-based methods of terahertz tuning rely on simply varying the repetition rate[9,31,49], which is inherently not operational when the source comb has to be under strict stabilization[50]. In contrast, without loosening the source comb stabilization, our terahertz synthesizer permits one comb line $\nu_1$ to hop over to the next comb line whereas the other comb line $\nu_2$ stays fixed (Fig. 4a). Specifically, the DFB laser diode conducting the injection-locking of $\nu_1$ is put on a programmable sequence of current control so that the operating window of injection-locking is shifted successively in steps of the repetition rate of 100 MHz (Fig. 4b). To validate the tunability of our synthesizer, two distinct scanning sequences are illustrated; one is a linearly up-down pattern over a 1.0 GHz range with ten stopovers of 10-s duration per each (Fig. 4c). The other is a complicated route programmed following

the letter logo of KAIST (Fig. 4d) over a tuning range of 2.0 GHz (details in "Methods").

## Overall performance comparison

Figure 5 shows how our terahertz synthesizer surpasses state-of-the-art counterparts in terms of the frequency stability vs. operating frequency range (Fig. 5a) and the phase noise level (Fig. 5b). As compared to quantum cascade lasers (QCLs)[21–24], heterodyne mixing cw lasers[31–35], and microcomb-based photomixing[36–38], our synthesizer provides improved short-term frequency stability over a wide operating range of 0.1–1.1 THz. Note that QCLs produce high-power terahertz waves spanning a 1–5 THz range with a wide tuning range over 1 THz, while their continuous tuning range has to be narrowly restricted when locked to a CW frequency reference[2,24]. Microcomb-based sources offer relatively higher stabilities over 0.1–1.5 THz, but their tuning capability is severely restricted due to the difficulty of cavity length regulation. In terms of the phase noise, our synthesizer shows superior performance at 1-Hz offset, lowered by 30 dB in comparison to the state-of-the-art counterparts, while QCLs injection-locked to a frequency comb[23] and stabilized microcombs[38] have achieved lower noise floors at offset frequencies above 100 Hz. The current noise floor of our synthesizer, being limited by the not-strong-enough power of terahertz waves emitted from the used photomixer coupled with the thermal noise level of the used PCA, will be further enhanced in the near future with the advent of high-power photomixers and low-noise PCAs.

## Discussion

The terahertz synthesizer demonstrated in this work is capable of providing a widened tuning range of 0.10–1.1 THz with a 2 mHz linewidth in stabilization to a high-finesse optical cavity made of ultra-low expansion glass. The frequency instability turns out to be $4.4 \times 10^{-15}$ at 1-s integration, with a two-order-of-magnitude improvement from the state-of-the-art counterparts relying on the radio-frequency or microwave standards. Further, the frequency instability reaches $5.1 \times 10^{-17}$ when the integration time is prolonged to 65 s, exhibiting the potential of down-converting the existing optical clock accuracy to the terahertz region. The maximum tuning range is currently set at 4 THz in consideration of the spectral bandwidth of the source comb, which can readily be extended to more than 10 THz by supercontinuum generation using a highly nonlinear fiber[51]. Meanwhile, the frequency noise is limited mainly by the signal-to-noise ratio of the used photomixer and PCA, which will be alleviated gradually in the near future with the advent of more high-power photomixers and high-sensitivity emitters. Nonetheless, the unprecedented frequency stability and accuracy achieved to a 15-digit level at 1-s integration, for the first time in the terahertz regime to our knowledge, will facilitate diverse advances in high precision terahertz spectroscopic metrology, low noise terahertz radar, and future wireless communications. Moreover, the current scheme of our synthesizer may be refined as a de facto industrial standard of terahertz frequency as done for the optical frequency synthesizer using photonic integration[52].

## Methods

### Comb line extraction

The optical frequency comb (OFC) used as the source comb in Fig. 1a is based on an Er-doped fiber femtosecond laser (C-fiber, Menlo Systems). The OFC offers a 60 nm wavelength bandwidth centered at 1550 nm, equivalent to a 4 THz frequency span, with a 100 MHz comb line-to-line spacing. The total comb power is set at 20 mW, while a single comb line is given ~200 nW. For each comb line to be extracted, a narrow-band spectral filter of a 100 MHz bandwidth (FWHM) is devised individually by combining a fiber Fabry-Perot (FFP) filter with a fiber Bragg grating (FBG) filter[39]. The filtered-out comb line is then injection-locked to a distributed feedback (DFB) laser diode for power

amplification, with a factor of 40–50 dB, without notable frequency shifting or linewidth degradation.

### Source comb stabilization

The optical cavity (ATF6300, Stable Laser Systems) shown in Fig. 1b is made of ultra-low expansion (ULE) glass, providing resonance peaks evenly repeated with a free spectral range (FSR) of 3.14 GHz. Each resonance peak offers a narrow bandwidth of 8 kHz (FWHM) with a high finesse of $4 \times 10^5$ under strict regulation of ambient temperature, humidity and vibration. The PDH control of locking one comb line to a nearby resonance peak is implemented with phase modulation using an electro-optic modulator (EOM) at a driving frequency of 110 MHz. The subsequent PDH error signal is detected using a high-speed photodetector through the locked-in amplification to the EOM driving frequency. The PDH error is then suppressed by activating an acousto-optic modulator (AOM) that is set to move the whole frequency structure of the source comb, up and down laterally, along the drive frequency symboled as $f_{AOM}$. Now let $f_o$ be the carrier-envelope offset of the source comb that is set to be adjusted by the current control of the pump LD connected to the source comb (Fig. 1b). The total offset of the source comb is the sum of $f_o$ and $f_{AOM}$ since the PDH control command also affects the overall offset. Then, the offset $f_o$ detected using an $f - 2f$ interferometer is controlled by a phase-locked loop to counterbalance $f_{AOM}$, i.e., $f_o = -f_{AOM}$. This leads the whole source comb to be stabilized to the ULE cavity with a zero carrier-envelope offset, i.e., $f_o + f_{AOM} = 0$. Consequently, each comb line settles at its corresponding optical frequency of $v_n = n \times f_r$ as an integer multiple of the repetition rate $f_r$.

### Terahertz generation and detection

For heterodyning photomixing, an InGaAs p-i-n photodiode-based photomixer (PCA-FD-1550-100-TX-1, Toptica) is used. The input optical power to the photomixer is set at 28 mW with a bias voltage of −0.8 V so as to produce an average photocurrent of 8 mA. For terahertz detection, a low-temperature InGaAs/InAlAs photoconductive antenna (TERA 15-RX-FC, Menlo Systems) is used. The gate pulse train incident to the photoconductive antenna (PCA) for the down-converted generation of a terahertz comb is adjusted to have an optical power of 30 mW with a pulse duration of 130 fs through dispersion compensation of the fiber delivery line. The photocurrent generated from the PCA is amplified by a transimpedance amplifier (TIA, LCA-200K-20M, Femto) to a voltage signal.

### Compensation for fiber thermal noise

In Fig. 2a, two comb lines ($v_1$ and $v_2$) to be heterodyned for are up-shifted by $f_{a1}$ (=39.95 MHz) and by $f_{a2}$ (=40.05 MHz), respectively. Then, each shifted comb line is beat with its original comb line, of which the RF Doppler signal corresponding to $f_{a1}$ and $f_{a2}$ is corrected through an AOM by PLL control for active compensation of the fiber thermal phase noise (Fig. 2b). At the same time, the optically carried terahertz defined by the optical difference of $|v_1 - v_2|$ yields an intermediate beatnote of 100 kHz, imposed by the difference of $f_{a2} - f_{a1}$, when beating with the original difference of the source comb. The RF beatnote is sampled using an FFT analyzer (SR760, SRS), from which the phase noise spectra of the optically carried terahertz signal $|v_1 - v_2|$ are evaluated accordingly (Fig. 2c).

### Terahertz phase noise measurement

The real terahertz wave of $|v_1 - v_2|$, radiated from the photomixer, is detected in the form of the same 100 kHz beatnote in interference with the terahertz comb down-converted from the source comb. This beatnote leads to the phase noise spectrum of the real 0.1 THz waves through an FFT analyzer (Fig. 2c). Note that, for a comparative purpose, the real 0.1 THz waves produced from the photomixer are processed directly using a commercial terahertz spectrum analyzer

(E4440A, Agilent) instead of referencing to the source comb using the PCA and TIA. The ready-made instrument operates with reference to the Rb clock through an RF-to-terahertz harmonic mixer (11970W, Agilent). The resulting phase noise measured from this spectrum analyzer is added to Fig. 2c (orange curve), which appears much larger than the phase noise measured through the PCA (red curve).

## Noise analysis in PCA heterodyne detection
The phase noise floor of the real terahertz signal (red curve, Fig. 2c) is contributed mainly by three distinct noise sources, each being distinguished as $N1$, $N2$ and $N3$. Note $N1$ represents the TIA's intrinsic noise, while $N2$ and $N3$ refer to the PCA's photocurrent shot noise and thermal (Johnson-Nyquist) noise, respectively. The 100 kHz beatnote signal delivered from the PCA carries an average photocurrent of 3.6 nA, producing a shot noise ($N2$) of 34 fA$\sqrt{\text{Hz}}$ with a phase noise amplitude of −100 dBc/Hz. Meanwhile, the thermal noise ($N3$) of the PCA and the intrinsic TIA noise ($N1$) are measured using an FFT analyzer (SR760, SRS). With the TIA's input placed in an open-circuit state with a transimpedance gain of $2 \times 10^7$ V/A, the TIA's intrinsic noise ($N1$) is found to be 35 fA/$\sqrt{\text{Hz}}$ at a 100 kHz offset, which causes a phase noise floor of −100 dBc/Hz. Next, with the PCA output being connected to the TIA input, the thermal noise ($N3$) of the PCA is monitored to be 560 fA/$\sqrt{\text{Hz}}$ which is predicted to be the most dominating noise floor of −76 dBc/Hz. The thermal noise ($N3$) shows no dependence on the optical pump power induced on the PCA in the 0–30 mW range, which is known as a typical character of the thermal noise created from the PCA[53]. The estimated thermal noise level of −76 dBc/Hz agrees well with the actually measured result of −73 dBc/Hz with a minimal unaccounted 3 dB discrepancy. Note that the PCA noise limit may also be mitigated by either increasing the repetition rate of the source comb or employing a sensitive Schottky diode detector instead of the PCA, which will be attempted in our near-future follow-up investigations.

## Phase-locked loop (PLL) of Comb #2
In Fig. 3a, for the PLL control of synchronization between the source comb and Comb #2, the beatnotes $f_{b1}$ and $f_{b2}$ between the comb lines $\nu_1$ and $\nu_2$ and Comb #2 are detected individually using photodetectors (PDA10CF-EC, Thorlabs). The beatnotes are then band-pass filtered, frequency-divided with a factor of 20, and mixed with the pre-assigned values given by the local oscillator. The resulting error signals in $f_{b1}$ and $f_{b2}$ are then individually PLL-corrected through PID servo controllers (FALC 110, Toptica) by the VCO feedback modulation driving the AOM as well as the pump current port configured in Comb#2 for its own stabilization.

## Measurements of linewidth and Allan deviation
In Fig. 3b, c, the heterodyned 100 Hz beat $f_{det}$ is observed up to 100 Hz using a double-balanced mixer (ZAD 3+, Mini-circuits), subsequently followed by a programmable band-pass filter (SR650, SRS) set to 50–150 Hz. Then the beat $f_{det}$ is recorded using an oscilloscope (DSO6012A, Agilent) for a sampling time of 500 s, with the linewidth being obtained by performing a fast Fourier transformation of 500,000 points with a sampling interval of 1 ms. This provides a frequency resolution of 2 mHz with a corresponding Nyquist frequency limit of 500 Hz. Next, the frequency stability of generated terahertz signals is evaluated in terms of the modified Allan deviation. For 0.10 and 0.66 THz signals, the measured 100 Hz beat $f_{det}$ is converted to the phase signal, $\phi(t)$, being compared to an ideal 100 Hz sinusoidal wave function. Then the instantaneous frequency is calculated as $f(t) = f_0 + (1/2\pi) \, d\phi(t)/dt$ and subsequently used to calculate Allan deviations with a gating time of 10 ms. On the other hand, the frequency stability of the 1.1 THz signals is measured using a spectrum analyzer (E4440A, Agilent) as it offers a relatively lower signal-to-noise ratio (20 dB at 1 Hz RBW) at a 200 ms gating time.

## Successive tuning of terahertz frequencies
In Fig. 4, in response to the input voltage, the DFB laser output frequency increases at a rate of 26.4 GHz/V, with mode hopping at every step of ~30 MHz. For the successive tuning of the terahertz frequency, the optical frequency comb is filtered by a FBG filter with a 1-nm transmission window and injected into the DFB laser. The ratio between the optical powers of the input comb line and the DFB laser is chosen to have an injection-locking range broader than the voltage step of mode hopping, and at the same time narrower than the repetition rate of 100 MHz of the source comb. This condition ensures only a single comb line is injection-locked with a signal-to-noise ratio (SNR) of 57.8 ± 3.2 dB and a side-mode suppression ratio (SMSR) of 29.7 ± 2.8 dB.

## Performance comparison
In Fig. 5a, the short-term stability of our synthesizer is compared to several state-of-the-art counterparts in terms of the fractional frequency instability (Allan deviation) at 1 s averaging. For those with no clearly-matching data highlighted in their literature, corresponding Allan deviations were estimated by calculation as marked with an asterisk in the figure−by dividing the spectral linewidth with the carrier frequency. Note that the spectral linewidth can be converted to the Allan deviation at a certain measurement sweeping time as both the quantities are interrelated through the frequency noise spectrum[54]. Detailed information such as the linewidths, resolution bandwidths, and measurement time are given in Table S1 in Supplementary Information.

## Data availability
The data that support the findings of this study are available from the corresponding author upon reasonable request.

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

## Acknowledgements

This work was financially supported by the National Research Foundation of the Republic of Korea (NRF-2012R1A3A1050386 and NRF-2019K1A3A1A20092429).

## Author contributions

This project was initiated and overseen by Y.-J.K. and S.-W.K. Comb stabilization to a high finesse cavity was prepared by D.-C.S. and B.S.K.

with the support of H.J. The experiments for terahertz frequency generation and detection were implemented, operated, and analyzed by D.-C.S. and B.S.K. The terahertz frequency tuning experiments were performed and analyzed by D.-C.S. All authors contributed to the manuscript preparation.

## Competing interests

The authors declare no competing interests.
