## [Peer review file · Nature Communications]

REVIEWER COMMENTS

Reviewer #1 (Remarks to the Author):

The manuscript titled “Optical cavity referenced terahertz synthesizer with 15 digits short term stability” demonstrates a photonic scheme of terahertz (THz) synthesis using an optical comb and claims ultra-high accuracy of their THz synthesizer. However, I find the paper to be engineering extensive and does not have any new physics. In the current form, the manuscript is not in a shape for publication in Nature communication.

Below are some comments for the authors to improve:

1. In the case of direct optical heterodyning of cw lasers through a photomixer like UTC-PD or pin-diode stable THz signals are generated by suppressing the phase noise. In their introduction, the authors have cited references 35-37 for this but have not highlighted why their method or approach is better than these. The authors must mention the limitations of existing state-of-the-art approaches and bring out their novelty. In conclusion, the introduction needs refinement and must compare their approach with existing approaches.
2. In the THz synthesizer an optical cavity is a crucial component. In this case, it is an ultra-low expansion (ULE) glass cavity which is one of the key elements as it provides the optical frequency comb. The authors should provide the details of the optical cavity such as its quality factor and the thermal stability of the cavity resonance.
3. The authors claim that the power of the filtered comb is amplified up to ~ 20 mW by injection locking to DFB laser. During this process, the original stability of the source comb does not degrade. The authors should provide experimental evidence for this.
4. The authors can further simplify their explanation of why they have chosen the Pound–Drever–Hall (PDH) technique.
5. As the frequency stability of the generated THz wave depends on the thermal noise of the fibre delivery line, the authors must justify their choice of using a 5m long fibre delivery line. How does the performance change with the length of the fibre delivery line?

6. Can the authors comment on how the total comb power was chosen to be 20mW? And how does the SSB Phase Noise Floor in Fig 2C vary with the injection power?

7. Can the authors explain why they have chosen to subtract 110,799,983,682,484 MHz in fig 1c and fig 4b to illustrate the THz frequency generation with 100 MHz steps? On a similar note, the authors must highlight how they arrive at the subtrahends on the y-axis of fig 4c and fig 4d.

8. Regarding the frequency tuning of the generated THz waves, the authors highlight that the existing comb-based methods rely on simply varying the repetition rate. To further substantiate their claim that this current scheme could be refined as a de-facto industrial standard for terahertz frequency synthesis, they should add a detailed comparison which also includes current commercial setups. For example, commercial THz TOPTICA systems provide a large frequency tuning of THz waves of up to 2.7 THz with excellent frequency resolutions of < 10 MHz.

Reviewer #2 (Remarks to the Author):

Dear Author,

The paper presents a THz source generated from an optical frequency comb and photomixing. The complex experiment is done very well and the text was easy to read. One of the things I really do not like about typical Nature papers is having to read the supplemental information to understand what is going on. I could do this from the text, and then the figures were great! I am pleased to say I am happy to accept the paper.

The one question I have is the fiber phase noise comes from splitting the comb to lock to the 2 separate LDs used to generate the THz. Is there anyway to make that common path?

Reviewer #3 (Remarks to the Author):

I have the following comments and concerns about the manuscript "Optical cavity-referenced terahertz synthesizer with 15-digit short-term stability" by Professor Kim and colleagues.

1) My first main concern is the novelty of this work. The main concept of the high-stability generation of the optical combs through the same Er-doped fiber oscillator used in this work (Setup shown in Figure 1) was previously demonstrated by the authors and published in Scientific Reports [Jang, et al. Comb-rooted multi-channel synthesis of ultra-narrow optical frequencies of few Hz linewidth. Scientific reports, 9 (2019)] and the impressive 10–15 stability was achieved. The authors converted this high-stability optical comb to terahertz using a commercially available photomixer from Toptica and since the stability of terahertz radiation is dictated by the stability of the optical lines, the high stability is maintained for the down-converted terahertz signal. Therefore, it is not clear to me what NEW knowledge and technical advancements are presented in this work.

2) My second main concern is how the authors benchmark their results in comparison with the state-of-the-art. Specifically, I question the comparisons shown in Figure 5. ‘Fractional frequency instability at 1s’ and ‘SSB phase noise’ of the authors’ results and other references [21-23] and [30-37] are listed in Figure 5. However, it is not clear how many of these parameters are extracted from these references. I personally looked at these cited references and could not see how many of these parameters are calculated. In addition, the comparisons are not thorough and do not include many of the prior art. While there are many examples to list here, one example is the reference [Freeman, et al. Injection locking of a terahertz quantum cascade laser to a telecommunications wavelength frequency comb. Optica, 4(9), pp.1059-1064 (2017)] that is not included, which demonstrates similar phase noise performance that we see in this manuscript, questioning a fair comparison with the prior art.

3) There is not enough experimental and theoretical support for the 2 mHz frequency resolution and synthesized linewidth. This should be clarified and the actual measured spectral linewidth before and after stabilization should be added to the experimental results.

4) The authors state that “Note that QCLs produce higher frequencies over 2 THz, but their tuning is currently limited to a sub-GHz range”, which is totally wrong. For example, VECSEL QCLs have provided more than 880 GHz tuning range as reported in this reference and other references [Curwen, et al. Broadband continuous single-mode tuning of a short-cavity quantum-cascade VECSEL. Nature Photonics, 13(12), pp.855-859 (2019)]. In addition, monolithic DFG-QCL tuners have demonstrated single-mode THz emission with a tuning range of 2.06–4.35 THz at room temperature [Lu et. al. Room temperature continuous wave, monolithic tunable THz sources based on highly efficient mid-infrared quantum cascade lasers. Sci Rep. 6:23595 (2016)].

5) The authors mention “Our terahertz synthesizer is also found superior in terms of the phase noise amplitude, particularly for lower offset frequencies below 1 kHz”. As mentioned above, I question a fair comparison of the authors’ results with the state of the art. Despite this, even with the results they are showing in Figure 5b, this statement is misleading since there are several other work that demonstrate better phase noise at offset frequencies above 1 kHz.

6) The authors should calculate and compare their frequency noise power spectral density (FNPSD) with the state of the art, which is a more insightful measure for a synthesizer.

7) The authors should discuss the physical factors that limits the frequency tuning range of their synthesizer. Especially, for spectroscopy applications, many of the spectral fingerprints are above 1 THz.

8) I suggest not using the term "15-digit accuracy" because the stability is very much affected by the integration time.

<Revision Summary>

Manuscript title: Optical cavity-referenced terahertz synthesizer with 15-digit short-term stability

[The reviewers' comments are reproduced in black for reference. The authors' responses are given in blue. Revised texts are quoted in *italic blue with indentation.*]

[Authors' response to Reviewer #1]

Reviewer #1's comment:

The manuscript titled "Optical cavity referenced terahertz synthesizer with 15 digits short term stability" demonstrates a photonic scheme of terahertz (THz) synthesis using an optical comb and claims ultra-high accuracy of their THz synthesizer. However, I find the paper to be engineering extensive and does not have any new physics. In the current form, the manuscript is not in a shape for publication in Nature communication. Below are some comments for the authors to improve:

1. In the case of direct optical heterodyning of cw lasers through a photomixer like UTC-PD or pin-diode stable THz signals are generated by suppressing the phase noise. In their introduction, the authors have cited references 35-37 for this but have not highlighted why their method or approach is better than these. The authors must mention the limitations of existing state-of-the-art approaches and bring out their novelty. In conclusion, the introduction needs refinement and must compare their approach with existing approaches.

In response to the reviewer's comment, the introduction part has been supplemented as;

(On page 2, line 22)

".... In terms of the frequency stability, the best value achieved so far reaches 2×10^{-13} at 1 s averaging, relying on a microwave H-maser³⁶. Nonetheless, there is room for better frequency references, such as a high-finesse optical cavity, to be employed in order to enhance the frequency stability of terahertz generation to be enough to transfer the state-of-the-art precision of optical clocks.

In this investigation, we present a practical scheme of frequency comb-based terahertz synthesizer, devised to transfer the superior frequency stability of a high-finesse optical cavity made of ultra-low expansion glass over a tunable range of 0.1 to 1.1 THz. Particular attention is paid to realizing coherent optical-to-terahertz down-conversion by heterodyne photomixing so as to reduce the linewidth as narrow as 2 mHz with a short-term frequency instability of about 10^{-15} at 1 s.

2. In the THz synthesizer an optical cavity is a crucial component. In this case, it is an ultra-low expansion (ULE) glass cavity which is one of the key elements as it provides the optical frequency comb. The authors should provide the details of the optical cavity such as its quality factor and the thermal stability of the cavity resonance.

As the reviewer suggested, more information has been given in the revised manuscript as;

(On page 3, line 19)

“ ... The high-finesse optical cavity provides narrow transmission lines of 8 kHz bandwidth (FWHM), corresponding to a finesse value of 4×10^5 or a quality factor of 2×10^{10} (See Methods for more details). ... ”

3. The authors claim that the power of the filtered comb is amplified up to ~ 20 mW by injection locking to DFB laser. During this process, the original stability of the source comb does not degrade. The authors should provide experimental evidence for this.

Theoretical and experimental evidence on the linewidth stability of the injection-locked comb lines are available in several papers of the authors' own including Ref. 39. In order to avoid duplication and provide more objective data, other groups' results of Ref. 43 and 44 have been added in the revised manuscript as;

(on page 4, line 8)

“The filtered comb lines are taken out, independently one by one, by injection locking to a distributed-feedback laser diode (DFB LD) permitting power amplification to ~ 20 mW without degradation of the original stability inherited from the source comb^{39,43,44}. ”

4. The authors can further simplify their explanation of why they have chosen the Pound–Drever–Hall (PDH) technique.

Among several alternatives, such as the Hänsch–Couillaud technique (Optics Communications, 35, 3, 1980), the validation of employing the PDH technique in our work has been simplified as;

(page 4, line 12)

“...which allows for a fast and robust stabilization with a large capture range⁴⁵ (Methods for details). As a consequence, the source comb is fully stabilized with the f_0 nullification, permitting the n -th comb line to settle to an optical frequency of $\nu_n = n \times f_r$ with a zero offset frequency^{39,46}”

5. As the frequency stability of the generated THz wave depends on the thermal noise of the fibre delivery line, the authors must justify their choice of using a 5m long fibre delivery line. How does the performance change with the length of the fibre delivery line?

This comment has been reflected in the revised manuscript as;

(page 4, line 19)

“ ... The fibre delivery line per each comb line requires a minimum length of 5 m to accommodate all the functional components—such as a FFP filter, a FBG filter and an

AOM—from the source comb to the photomixer. The lengthy fibre delivery causes a fractional frequency instability of $\sim 10^{-16}$ at 1–100 s averaging when monitored at the inlet of the photomixer by optical beating either v_1 or v_2 with its original comb line of the source comb (Fig. 2b)."

6. Can the authors comment on how the total comb power was chosen to be 20mW? And how does the SSB Phase Noise Floor in Fig 2C vary with the injection power?

The 20 mW power of each comb line corresponds to the normal operating power of the DFB laser used as the slave laser for injection locking. At the moment, we have no experimental data showing how the SSB phase noise floor in Fig. 2c would vary with the injection power, particularly when it is lowered far below 20 mW.

7. Can the authors explain why they have chosen to subtract 110,799,983,682,484 MHz in fig 1c and fig 4b to illustrate the THz frequency generation with 100 MHz steps? On a similar note, the authors must highlight how they arrive at the subtrahends on the y-axis of fig 4c and fig 4d.

(on page 7, line 21)

"Note that the subtrahend on the y-axis of Fig 4c and Fig 4d is the terahertz frequency f_i determined before initiating the successive tuning; e.g., with the absolute mode number, $n_i = 1108$, and the comb repetition rate, $f_c = 99.999985273$ MHz, the initial f_i is calculated to be 110.799983682484 GHz."

8. Regarding the frequency tuning of the generated THz waves, the authors highlight that the existing comb-based methods rely on simply varying the repetition rate. To further substantiate their claim that this current scheme could be refined as a de-facto industrial standard for terahertz frequency synthesis, they should add a detailed comparison which also includes current commercial setups. For example, commercial THz TOPTICA systems provide a large frequency tuning of THz waves of up to 2.7 THz with excellent frequency resolutions of < 10 MHz.

We appreciate the reviewer's suggestion. As pointed out, our comparison is currently restricted to comb-related terahertz synthesizers capable of identifying their absolute frequencies in real time with reference to the atomic clocks or optical references. More comprehensive comparisons including state-of-the-art commercial products will be considered in our next work.

[Authors' response to Reviewer #2]

Reviewer #2's comment:

The paper presents a THz source generated from an optical frequency comb and photomixing. The complex experiment is done very well and the text was easy to read. One of the things I really do not like about typical Nature papers is having to read the supplemental information to understand what is going on. I could do this from the text, and then the figures were great! **I am pleased to say I am happy to accept the paper.**

The one question I have is the fiber phase noise comes from splitting the comb to lock to the 2 separate LDs used to generate the THz. Is there anyway to make that common path?

We very much appreciate the reviewer's suggestion. Our next work will be focused on extracting multiple comb lines through a common-path fiber line with the capability of inherent subtraction of thermal noise.

[Authors' response to Reviewer #3]

Reviewer #3's comment:

I have the following comments and concerns about the manuscript "Optical cavity-referenced terahertz synthesizer with 15-digit short-term stability" by Professor Kim and colleagues.

1) My first main concern is the novelty of this work. The main concept of the high-stability generation of the optical combs through the same Er-doped fiber oscillator used in this work (Setup shown in Figure1) was previously demonstrated by the authors and published in Scientific Reports [Jang, et al. Comb-rooted multi-channel synthesis of ultra-narrow optical frequencies of few Hz linewidth. Scientific reports, 9 (2019)] and the impressive 10^{-15} stability was achieved. The authors converted this high-stability optical comb to terahertz using a commercially available photomixer from Toptica and since the stability of terahertz radiation is dictated by the stability of the optical lines, the high stability is maintained for the down-converted terahertz signal. Therefore, it is not clear to me what NEW knowledge and technical advancements are presented in this work.

In consideration of the reviewer's concern, relevant discussions have been supplemented in the introduction and conclusion parts of the revised manuscript as;

(On page 2, line 22)

".... In terms of the frequency stability, the best value achieved so far reaches 2×10^{-13} at 1 s averaging, relying on a microwave H-maser³⁶. Nonetheless, there is room for better frequency references, such as a high-finesse optical cavity, to be employed in order to enhance the frequency stability of terahertz generation to be enough to transfer the state-of-the-art precision of optical clocks.

In this investigation, we present a practical scheme of frequency comb-based terahertz synthesizer, devised to transfer the superior frequency stability of a high-finesse optical cavity made of ultra-low expansion glass over a tunable range of 0.1 to 1.1 THz. Particular attention is paid to realizing coherent optical-to-terahertz down-conversion by heterodyne photomixing so as to reduce the linewidth as narrow as 2 mHz with a short-term frequency instability of about 10^{-15} at 1 s. With the unprecedented performance, our terahertz synthesizer is intended to accelerate terahertz applications for extremely low noise radar, 6G wireless communications, and high resolution scientific molecular spectroscopy.

(On page 8, line 19)

" The terahertz synthesizer demonstrated in this work is capable of providing a widened tuning range of 0.10 – 1.1 THz with a 2 mHz linewidth in stabilization to a high-finesse optical cavity made of ultra-low expansion glass. The short-term frequency instability reaches 4.4×10^{-15} at 1 s integration, with a two-order-of-magnitude improvement from the state-of-the-art counterparts relying on the radio-frequency or microwave standards. Further, the frequency instability reduces to 5.1×10^{-17} when the integration is prolonged to 65 s, exhibiting the potential of being capable of down-converting the existing optical clock accuracy to the terahertz region. Currently, overall performance, particularly in terms of the maximum tuning range and frequency noise, is limited mainly by the signal-

to-noise ratio of the used photomixer and PCA, which will be alleviated gradually in the near future with the advent of more high-power photomixers and high-sensitivity emitters. Nonetheless, the unprecedented frequency stability and accuracy achieved to a 15-digit level at 1 s integration, for the first time in the terahertz regime to our knowledge, will facilitate diverse advances in high precision terahertz spectroscopic metrology, low noise terahertz radar, and future wireless communications. Moreover, the current scheme of our synthesizer may be refined as a de-facto industrial standard of terahertz frequency as done for the optical frequency synthesizer using photonic integration⁵².

2) My second main concern is how the authors benchmark their results in comparison with the state-of-the art. Specifically, I question the comparisons shown in Figure 5. ‘Fractional frequency instability at 1s’ and ‘SSB phase noise’ of the authors’ results and other references [21-23] and [30-37] are listed in Figure 5. However, it is not clear how many of these parameters are extracted from these references. I personally looked at these cited references and could not see how many of these parameters are calculated. In addition, the comparisons are not thorough and do not include many of the prior art. While there are many examples to list here, one example is the reference [Freeman, et al. Injection locking of a terahertz quantum cascade laser to a telecommunications wavelength frequency comb. *Optica*, 4(9), pp.1059-1064 (2017)] that is not included, which demonstrates similar phase noise performance that we see in this manuscript, questioning a fair comparison with the prior art.

As for the clarification on how the fractional frequency instability is extracted, relevant explanations have been given in Methods in the revised manuscript as;

(on page 13, line 20)

“Performance comparison. In Fig. 5a, the short-term stability of our synthesizer is compared to several state-of-the-art counterparts in terms of the fractional frequency instability (Allan deviation) at 1 s averaging. For those with no clearly-matching data highlighted in their literature, corresponding Allan deviations were estimated by calculation as marked with an asterisk in the figure - by dividing the spectral linewidth with the carrier frequency. Note that the spectral linewidth can be converted to the Allan deviation at a certain measurement sweeping time as both the quantities are interrelated through the frequency noise spectrum⁵³.”

53. Turner, L. ., Weber, K. ., Hawthorn, C. . & Scholten, R. . Frequency noise characterisation of narrow linewidth diode lasers. *Opt. Commun.* 201, 391–397 (2002).

3) There is not enough experimental and theoretical support for the 2 mHz frequency resolution and synthesized linewidth. This should be clarified and the actual measured spectral linewidth before and after stabilization should be added to the experimental results.

As far as the linewidth stability of the injection-locked comb lines is concerned, extensive theoretical and experimental evidence has been published in several papers of the authors’ own

including Ref. 39. In order to avoid duplication and provide more objective data, other groups' results of Ref. 43 and 44 have been added in the revised manuscript as;

(on page 4, line 8)

“The filtered comb lines are taken out, independently one by one, by injection locking to a distributed-feedback laser diode (DFB LD) permitting power amplification to ~ 20 mW without degradation of the original stability inherited from the source comb^{39,43,44}.”

4) The authors state that “Note that QCLs produce higher frequencies over 2 THz, but their tuning is currently limited to a sub-GHz range”, which is totally wrong. For example, VECSEL QCLs have provided more than 880 GHz tuning range as reported in this reference and other references [Curwen, et. al. Broadband continuous single-mode tuning of a short-cavity quantum-cascade VECSEL. Nature Photonics, 13(12), pp.855-859 (2019)]. In addition, monolithic DFG-QCL tuners have demonstrated single-mode THz emission with a tuning range of 2.06–4.35 THz at room temperature [Lu et. al. Room temperature continuous wave, monolithic tunable THz sources based on highly efficient mid-infrared quantum cascade lasers. Sci Rep. 6:23595 (2016)].

Our misleading statement has been rephrased in the revised manuscript as;

(page 8, line 7)

“Note that QCLs produce high-power terahertz waves covering a 1–5 THz range with a wide tuning range over 1 THz, while their continuously tuning range has to be narrowly restricted when locked to a CW frequency reference^{2,24}.”

5) The authors mention “Our terahertz synthesizer is also found superior in terms of the phase noise amplitude, particularly for lower offset frequencies below 1 kHz”. As mentioned above, I question a fair comparison of the authors' results with the state of the art. Despite this, even with the results they are showing in Figure 5b, this statement is misleading since there are several other work that demonstrate better phase noise at offset frequencies above 1 kHz.

In consideration of the reviewer's comment, we have added a reference (Freeman, et al. Optica, 4, 9, 1059, 2017) and modified the sentence for fair comparison in the revised manuscript as;

(page 8, line 11)

“In terms of the phase noise, our synthesizer shows superior performance at 1 Hz offset, lowered by 30 dB in comparison to the state-of-the-art counterparts, while QCLs injection-locked to a frequency comb²³ and stabilized microcombs³⁸ have achieved lower noise floors at offset frequencies above 100 Hz. The current noise floor of our synthesizer, being limited by the emitted power of terahertz waves and the thermal noise of the PCA, will be further improved in the near future with the advent of high-power photomixers and low-noise PCAs.”

6) The authors should calculate and compare their frequency noise power spectral density (FNPSD) with the state of the art, which is a more insightful measure for a synthesizer.

We agree with the reviewer's comment, but most of the state-of-the-art terahertz sources referenced in our work provide no detailed data on FNPSD, not enabling us to perform an immediate comparison.

7) The authors should discuss the physical factors that limits the frequency tuning range of their synthesizer. Especially, for spectroscopy applications, many of the spectral fingerprints are above 1 THz.

The reviewer's suggestion has been reflected in the revised manuscript as;

(page 8, line 19)

“ The terahertz synthesizer demonstrated in this work is capable of providing a widened tuning range of 0.10 – 1.1 THz with a 2 mHz linewidth in stabilization to a high-finesse optical cavity made of ultra-low expansion glass. The short-term frequency instability reaches 4.4×10^{-15} at 1 s integration, with a two-order-of-magnitude improvement from the state-of-the-art counterparts relying on the radio-frequency or microwave standards. Further, the frequency instability reduces to 5.1×10^{-17} when the integration is prolonged to 65 s, exhibiting the potential of being capable of down-converting the existing optical clock accuracy to the terahertz region. Currently, overall performance, particularly in terms of the maximum tuning range and frequency noise, is limited mainly by the signal-to-noise ratio of the used photomixer and PCA, which will be alleviated gradually in the near future with the advent of more high-power photomixers and high-sensitivity emitters. Nonetheless, the unprecedented frequency stability and accuracy achieved to a 15-digit level at 1 s integration, for the first time in the terahertz regime to our knowledge, will facilitate diverse advances in high precision terahertz spectroscopic metrology, low noise terahertz radar, and future wireless communications. Moreover, the current scheme of our synthesizer may be refined as a de-facto industrial standard of terahertz frequency as done for the optical frequency synthesizer using photonic integration⁵². ”

8) I suggest not using the term “15-digit accuracy” because the stability is very much affected by the integration time.

In acceptance of the reviewer's suggestion, the term “15-digit accuracy” has been rephrased as “15-digit short-term stability”.

REVIEWER COMMENTS

Reviewer #1 (Remarks to the Author):

In the revised manuscript, the authors have addressed all my comments. However, I would suggest the authors include the metric of phase noise in addition to frequency stability in the introduction/ motivation part. In addition, the authors must provide a potential outlook to reduce further the phase noise arising from splitting the fiber line. Furthermore, the authors must include a brief explanation of the mechanism of how after stabilizing the source frequency comb to the ULE cavity, the exact precision of the frequency comb is translated to the generated terahertz signals and what could be the potential limiting factor?

In conclusion, this work opens up new avenues for developing stable terahertz sources which will advance the high-precision terahertz applications such as spectroscopy, imaging, and communication. After including the minor revisions suggested above, the paper deserves acceptance at Nature communication.

Reviewer #2 (Remarks to the Author):

Dear Authors,

Thank you for your reply to the reviewer comments. Unlike Reviewer #1 and #3 who make weak claims for "not novel," and "no new physics" I appreciate the engineering efforts you have made to transfer the stability of your optical comb to your THz beam and obtain a new record. These comments are just lazy/arbitrary ways to reject a great paper.

Reviewer #3 (Remarks to the Author):

As explained below, the authors have not even nearly addressed the serious concerns of this reviewer. I am reiterating my concerns in red. Based on the concerns below, I do not recommend publication of this manuscript in Nature Communications.

Reviewer #3's comment:

I have the following comments and concerns about the manuscript "Optical cavity-referenced terahertz synthesizer with 15-digit short-term stability" by Professor Kim and colleagues.

1) My first main concern is the novelty of this work. The main concept of the high-stability generation of the optical combs through the same Er-doped fiber oscillator used in this work (Setup shown in Figure1) was previously demonstrated by the authors and published in Scientific Reports [Jang, et al. Comb-rooted multi-channel synthesis of ultra-narrow optical frequencies of few Hz linewidth. Scientific reports, 9 (2019)] and the impressive 10^{-15} stability was achieved. The authors converted this high-stability optical comb to terahertz using a commercially available photomixer from Toptica and since the stability of terahertz radiation is dictated by the stability of the optical lines, the high stability is maintained for the down-converted terahertz signal. Therefore, it is not clear to me what NEW knowledge and technical advancements are presented in this work.

In consideration of the reviewer's concern, relevant discussions have been supplemented in the introduction and conclusion parts of the revised manuscript as;

(On page 2, line 22)

".... In terms of the frequency stability, the best value achieved so far reaches 2×10^{-13} at 1 s averaging, relying on a microwave H-maser³⁶. Nonetheless, there is room for better frequency references, such as a high-finesse optical cavity, to be employed in order to enhance the frequency stability of terahertz generation to be enough to transfer the state-of-the-art precision of optical clocks.

In this investigation, we present a practical scheme of frequency comb-based terahertz synthesizer, devised to transfer the superior frequency stability of a high-finesse optical cavity made of ultra-low expansion glass over a tunable range of 0.1 to 1.1 THz. Particular attention is paid to realizing coherent optical-to-terahertz down-conversion by heterodyne photomixing so as to reduce the linewidth as narrow as 2 mHz with a short-term frequency instability of about 10^{-15} at 1 s. With the unprecedented performance, our terahertz synthesizer is intended to accelerate terahertz applications for extremely low noise radar, 6G wireless communications, and high resolution scientific molecular spectroscopy.

(On page 8, line 19)

" The terahertz synthesizer demonstrated in this work is capable of providing a widened tuning range of 0.10 – 1.1 THz with a 2 mHz linewidth in stabilization to a high-finesse optical cavity made of ultra-low expansion glass. The short-term frequency instability reaches 4.4×10^{-15} at 1 s integration, with a two-order-of-magnitude improvement from the state-of-the-art counterparts relying on the radio-frequency or microwave standards. Further, the frequency instability reduces to 5.1×10^{-17} when the integration is prolonged to 65 s, exhibiting the potential of being capable of down-converting the existing optical clock accuracy to the terahertz region. Currently, overall performance, particularly in terms of the maximum tuning range and frequency noise, is limited mainly by the signal- to-

noise ratio of the used photomixer and PCA, which will be alleviated gradually in the near future with the advent of more high-power photomixers and high-sensitivity emitters. Nonetheless, the unprecedented frequency stability and accuracy achieved to a 15-digit level at 1 s integration, for the first time in the terahertz regime to our knowledge, will facilitate diverse advances in high precision terahertz spectroscopic metrology, low noise terahertz radar, and future wireless communications. Moreover, the current scheme of our synthesizer may be refined as a de-facto industrial standard of terahertz frequency as done for the optical frequency synthesizer using photonic integration⁵².

The discussions that the authors added are not relevant to the raised concern. In this manuscript, the authors present the same work they published in Scientific Reports [Jang, et al. Comb-rooted multi-channel synthesis of ultra-narrow optical frequencies of few Hz linewidth. Scientific reports, 9 (2019)]. In their prior publication, they used exactly the same setup and concept shown in Figure 1 and they achieved the same optical stability of 10^{-15} . Their current manuscript only added a commercially available photomixer from Toptica for frequency conversion to terahertz and it is no surprise that the same optical stability was achieved. A simple autocorrelation calculation can prove that the optical stability will be transferred to terahertz stability as previously demonstrated both experimentally and theoretically. I guess the fact that the authors did not address this concern means that they agree the optical setup shown in Figure 1 is the same as the one in their Scientific reports paper. The remaining is just engineering work, pumping a commercially available photomixer from Toptica with the optical beam. Therefore, as also stated by the first reviewer, I do not see new physics and knowledge presented in this manuscript.

2) My second main concern is how the authors benchmark their results in comparison with the state-of-the art. Specifically, I question the comparisons shown in Figure 5. ‘Fractional frequency instability at 1s’ and ‘SSB phase noise’ of the authors’ results and other references [21-23] and [30-37] are listed in Figure 5. However, it is not clear how many of these parameters are extracted from these references. I personally looked at these cited references and could not see how many of these parameters are calculated. In addition, the comparisons are not thorough and do not include many of the prior art. While there are many examples to list here, one example is the reference [Freeman, et al. Injection locking of a terahertz quantum cascade laser to a telecommunications wavelength frequency comb. Optica, 4(9), pp.1059-1064 (2017)] that is not included, which demonstrates similar phase noise performance that we see in this manuscript, questioning a fair comparison with the prior art.

As for the clarification on how the fractional frequency instability is extracted, relevant explanations have been given in Methods in the revised manuscript as;

(on page 13, line 20)

“Performance comparison. In Fig. 5a, the short-term stability of our synthesizer is compared to several state-of-the-art counterparts in terms of the fractional frequency instability (Allan deviation) at 1 s averaging. For those with no clearly-matching data highlighted in their literature, corresponding Allan deviations were estimated by calculation as marked with an asterisk in the figure - by dividing the spectral linewidth with the carrier frequency. Note that the spectral linewidth can be converted to the Allan deviation at a certain measurement sweeping time as both the quantities are interrelated through the frequency noise spectrum⁵³.”

53. Turner, L. ., Weber, K. ., Hawthorn, C. . & Scholten, R. . Frequency noise characterisation of narrow linewidth diode lasers. *Opt. Commun.* 201, 391–397 (2002).

The additional explanations that the authors added do not answer the concern raised by this reviewer. First, the authors totally ignored my question about how they calculated SSB phase noise. Second, the approach they described for fractional frequency instability does not lead to the results they are showing in Figure 5. This is a serious concern since Figure 5 benchmarks the authors result relative to the state of the art and the listed parameters are not from the cited references and not reproducible based on the authors explanation. For each of the data points included in Figure 5 (both parts a and b), the authors should justify the values by their calculations. Detailed calculations should be given in a supplementary file with all the parameters taken from these references included. In providing this information the measurement parameters such as RBW, integration time, measurement time, exact definition of linewidth, etc which directly impact the results should be clear and taken into account to provide a fair comparison. If there are two approaches taken for calculating fractional frequency instability, the authors should apply both approaches to their own results and prove with numbers that they are equal for their own data.

3) There is not enough experimental and theoretical support for the 2 mHz frequency resolution and synthesized linewidth. This should be clarified and the actual measured spectral linewidth before and after stabilization should be added to the experimental results.

As far as the linewidth stability of the injection-locked comb lines is concerned, extensive theoretical and experimental evidence has been published in several papers of the authors' own including Ref. 39. In order to avoid duplication and provide more objective data, other groups' results of Ref. 43 and 44 have been added in the revised manuscript as;

(on page 4, line 8)

“The filtered comb lines are taken out, independently one by one, by injection locking to a distributed-feedback laser diode (DFB LD) permitting power amplification to ~ 20 mW without degradation of the original stability inherited from the source comb^{39,43,44}.”

This response is unacceptable. The authors report a 2 mHz frequency resolution and synthesized linewidth as one of the main specs of their synthesizer. They cannot just mention the measurement method and state the final calculated value. They should include all the measurements and calculations done at 0.1, 0.6, and 1 THz that led to this 2 mHz linewidth calculation. They can put all the details in a supplementary file. They should compare the same measurements and calculations before and after stabilization.

4) The authors state that “Note that QCLs produce higher frequencies over 2 THz, but their tuning is currently limited to a sub-GHz range”, which is totally wrong. For example, VECSEL QCLs have provided more than 880 GHz tuning range as reported in this reference and other references [Curwen, et. al. Broadband continuous single-mode tuning of a short-cavity quantum- cascade VECSEL. *Nature Photonics*, 13(12), pp.855-859 (2019)]. In addition, monolithic DFG- QCL tuners have demonstrated single-mode THz emission with a tuning range of 2.06–4.35 THz at room temperature [Lu et. al. Room temperature continuous wave, monolithic tunable THz sources based on highly efficient mid-infrared quantum cascade lasers. *Sci Rep.* 6:23595 (2016)].

Our misleading statement has been rephrased in the revised manuscript as;

(page 8, line 7)

“Note that QCLs produce high-power terahertz waves covering a 1–5 THz range with a wide tuning range over 1 THz, while their continuously tuning range has to be narrowly restricted when locked to a CW frequency reference^{2,24}.”

5) The authors mention “Our terahertz synthesizer is also found superior in terms of the phase noise amplitude, particularly for lower offset frequencies below 1 kHz”. As mentioned above, I question a fair comparison of the authors’ results with the state of the art. Despite this, even with the results they are showing in Figure 5b, this statement is misleading since there are several other work that demonstrate better phase noise at offset frequencies above 1 kHz.

In consideration of the reviewer’s comment, we have added a reference (Freeman, et al. Optica, 4, 9, 1059, 2017) and modified the sentence for fair comparison in the revised manuscript as;

(page 8, line 11)

“In terms of the phase noise, our synthesizer shows superior performance at 1 Hz offset, lowered by 30 dB in comparison to the state-of-the-art counterparts, while QCLs injection-locked to a frequency comb²³ and stabilized microcombs³⁸ have achieved lower noise floors at offset frequencies above 100 Hz. The current noise floor of our synthesizer, being limited by the emitted power of terahertz waves and the thermal noise of the PCA, will be further improved in the near future with the advent of high-power photomixers and low-noise PCAs.”

The statement *“The current noise floor of our synthesizer, being limited by the emitted power of terahertz waves and the thermal noise of the PCA, will be further improved in the near future with the advent of high-power photomixers and low-noise PCAs.”* is misleading. The authors are using commercially available photomixers and PCAs like what many other references have used. Any advances in the radiation power and noise performance of commercially available photomixers and PCAs would have the same impact on all synthesizers demonstrated by the authors and other groups since all of them are based on the same frequency conversion process. This statement suggests that only the authors’ synthesizer would benefit from higher performance photomixers and PCAs, which is not true.

6) The authors should calculate and compare their frequency noise power spectral density (FNPSD) with the state of the art, which is a more insightful measure for a synthesizer.

We agree with the reviewer’s comment, but most of the state-of-the-art terahertz sources referenced in our work provide no detailed data on FNPSD, not enabling us to perform an immediate comparison.

The parameters that authors have used to benchmark their results (those shown in Figure 5) are not used by most of the references they have listed (as authors also agree and mention in their response to my second comment). Since the authors calculate the fractional frequency stability and SSB phase noise for other references, they can also calculate frequency noise power spectral density (FNPSD). In addition, it is very important to calculate the frequency noise power spectral density (FNPSD) of the demonstrated synthesizer as it is a better benchmark for stability of the

synthesizer.

7) The authors should discuss the physical factors that limits the frequency tuning range of their synthesizer. Especially, for spectroscopy applications, many of the spectral fingerprints are above 1 THz.

The reviewer's suggestion has been reflected in the revised manuscript as;

(page 8, line 19)

“ The terahertz synthesizer demonstrated in this work is capable of providing a widened tuning range of 0.10 – 1.1 THz with a 2 mHz linewidth in stabilization to a high-finesse optical cavity made of ultra-low expansion glass. The short-term frequency instability reaches 4.4×10^{-15} at 1 s integration, with a two-order-of-magnitude improvement from the state-of-the-art counterparts relying on the radio-frequency or microwave standards. Further, the frequency instability reduces to 5.1×10^{-17} when the integration is prolonged to 65 s, exhibiting the potential of being capable of down-converting the existing optical clock accuracy to the terahertz region. Currently, overall performance, particularly in terms of the maximum tuning range and frequency noise, is limited mainly by the signal- to-noise ratio of the used photomixer and PCA, which will be alleviated gradually in the near future with the advent of more high-power photomixers and high-sensitivity emitters. Nonetheless, the unprecedented frequency stability and accuracy achieved to a 15-digit level at 1 s integration, for the first time in the terahertz regime to our knowledge, will facilitate diverse advances in high precision terahertz spectroscopic metrology, low noise terahertz radar, and future wireless communications. Moreover, the current scheme of our synthesizer may be refined as a de-facto industrial standard of terahertz frequency as done for the optical frequency synthesizer using photonic integration⁵². ”

The authors say *“overall performance, particularly in terms of the maximum tuning range and frequency noise, is limited mainly by the signal- to-noise ratio of the used photomixer and PCA”* however, both photomixers and PCAs today have much larger tuning range than 1 THz (more than 5 THz). The authors should discuss the physical limitations of the optical combs generated by their presented scheme in providing higher terahertz frequencies.

8) I suggest not using the term “15-digit accuracy” because the stability is very much affected by the integration time.

In acceptance of the reviewer's suggestion, the term “15-digit accuracy” has been rephrased as “15-digit short-term stability”.

‘15-digit short term stability’ is very vague in describing the actual physical performance. The authors should mention the exact numbers in terms of both accuracy and time: for example, 4.4×10^{-15} at 1 s integration.

[Authors' responses to Reviewer #1]

In the revised manuscript, the authors have addressed all my comments. However, I would suggest the authors include (1) the metric of phase noise in addition to frequency stability in the introduction/motivation part. In addition, the authors must provide (2) a potential outlook to reduce further the phase noise arising from splitting the fiber line. Furthermore, the authors must include (3) a brief explanation of the mechanism of how after stabilizing the source frequency comb to the ULE cavity, the exact precision of the frequency comb is translated to the generated terahertz signals and (4) what could be the potential limiting factor?

In conclusion, this work opens up new avenues for developing stable terahertz sources which will advance high-precision terahertz applications such as spectroscopy, imaging, and communication. After including the minor revisions suggested above, **the paper deserves acceptance at Nature Communication.**

(1) Accepting the reviewer's suggestion, the metric of the phase noise in addition to the frequency stability has been included in the introduction part of our revised manuscript:

(on page 2, line 21)

“The best level of phase noise and frequency stability achieved yet reached -40 dBc/Hz at 1 Hz and 2×10^{-13} at 1 s, respectively, which were obtained by relying on the microwave H-maser³⁶.”

(2) A potential outlook to reduce further the fiber thermal noise has been added:

(on page 6, line 1)

“This result indicates a 30-dB improvement achieved by the PLL-controlled suppression of the fibre thermal noise (blue curve, Fig. 2c). The fibre thermal noise would be further reduced by minimizing the split length of the fibre lines needed for comb line extractions together with more rigorous regulation of the ambient temperature drift.”

(3) This suggestion has been reflected in the revised manuscript:

(on page 1, line 10)

“The source comb is an erbium-doped fibre oscillator locked to an ultra-low expansion optical cavity with subsequent $f-2f$ self-referencing, with all comb lines stabilized to offer a fractional frequency instability of 10^{-15} at 1-s integration. Then, pairs of comb lines are extracted from the source comb by injection locking to laser diodes without stability deterioration, which permits tunable terahertz generation by photomixing in incremental steps of 100 MHz over the range of 0.10 – 1.10 THz. Emphasis is put on the active compensation of fibre thermal noise to generate terahertz waves with an extremely low level of phase noise of -70 dBc/Hz at 1-Hz offset. The frequency instability measured for a 0.66 THz wave is 4.4×10^{-15} at 1-s integration, which reduces to 5.1×10^{-17} at 65-s integration.”

4) The potential limiting factor has been discussed in the revised manuscript at three locations;

(on page 1, line 18)

“The random white noise of the photoconductive antenna used for terahertz detection is found to be the dominating limiting factor to the delivery of the state-of-the-art optical clock stability down to the terahertz region over a bridge of three orders of frequency reduction.”

(on page 5, line 4)

“It is important to note that the frequency stability of the terahertz waves generated by photomixing is found affected by not only the source comb stability but also two extra causes; one is the thermal noise of the fibre delivery line and the other the conversion noise of the photomixer.”

(on page 9, line 14)

“...the frequency noise is limited mainly by the signal-to-noise ratio of the used photomixer and PCA, which will be alleviated gradually in the near future with the advent of more high-power photomixers and high-sensitivity emitters.”

[Authors' response to Reviewer #2]

Thank you for your reply to the reviewer's comments. Unlike Reviewers #1 and #3 who make weak claims for "not novel," and "no new physics" I appreciate the engineering efforts you have made to transfer the stability of your optical comb to your THz beam and obtain a new record. These comments are just lazy/arbitrary ways to reject a great paper.

The authors very much appreciate the reviewer's positive evaluation.

[Authors' responses to Reviewer #3]

I have the following comments and concerns about the manuscript "Optical cavity-referenced terahertz synthesizer with 15-digit short-term stability" by Professor Kim and colleagues.

1) The discussions that the authors added are not relevant to the raised concern. In this manuscript, the authors present the same work they published in Scientific Reports [Jang, et al. Comb-rooted multichannel synthesis of ultra-narrow optical frequencies of few Hz linewidth. Scientific reports, 9 (2019)]. In their prior publication, they used exactly the same setup and concept shown in Figure 1 and they achieved the same optical stability of 10^{-15} . Their current manuscript only added a commercially available photomixer from Toptica for frequency conversion to terahertz and it is no surprise that the same optical stability was achieved. A simple autocorrelation calculation can prove that the optical stability will be transferred to terahertz stability as previously demonstrated both experimentally and theoretically. I guess the fact that the authors did not address this concern means that they agree the optical setup shown in Figure 1 is the same as the one in their Scientific reports paper. The remaining is just engineering work, pumping a commercially available photomixer from Toptica with the optical beam. Therefore, as also stated by the first reviewer, I do not see new physics and knowledge presented in this manuscript.

Coherent frequency down-conversion of optical frequencies to lower frequencies is not a simple engineering work as it involves a series of identification/compensation tasks of uncommon path noises and other device-related noises through a multitude of phase-locked loops. Excellent examples are coherent optical-to-microwave conversions (*Fortier, T. et al.*, Nature Photon 5, 425–429 (2011) and *Xie, X. et al.*, Nature Photon 11, 44–47 (2017) as posted in Fig. RR1 for reference. These previous works used a cavity-stabilized frequency comb to generate ultra-stable microwave signals (10 GHz), in which the frequency comb stabilization was not completely novel. Nonetheless, they were recognized as meaningful demonstrations of ultra-stable microwave generation permitting diverse applications.

In this context, our work was intended to provide a framework for coherent optical-to-terahertz frequency down-conversion, with a novel scheme of uncommon-path noise compensation followed by verification of the residual noise level caused by the photomixer and photo-conductive antenna. As a result, the stability of the high-finesse optical cavity was transferred to the terahertz domain with a fractional frequency instability of 4.4×10^{-15} at 1-s integration, which is considered the most stable and precise terahertz waves demonstrated ever. Further, our comb line extraction technique allows for a novel method of tunable synthesis of ultra-stable terahertz waves over a wide spectral range, paving the way for diverse high-precision terahertz applications.

Generation of ultrastable microwaves via optical frequency division

T. M. Fortier*, M. S. Kirchner, F. Quinlan, J. Taylor, J. C. Bergquist, T. Rosenband, N. Lemke, A. Ludlow, Y. Jiang, C. W. Oates and S. A. Diddams*

Photonic microwave signals with zeptosecond-level absolute timing noise

Xiaopeng Xie¹, Romain Bouchand¹, Daniele Nicolodi¹, Michele Giunta^{2,3}, Wolfgang Hänsel², Matthias Lezius², Abhay Joshi⁴, Shubhashish Datta⁴, Christophe Alexandre⁵, Michel Lours¹, Pierre-Alain Tremblin⁶, Giorgio Santarelli⁶, Ronald Holzwarth^{2,3} and Yann Le Coq^{1*}

Fig. RR1 Examples of coherent down-conversion of optical comb frequencies to microwaves.

2-1) The additional explanations that the authors added do not answer the concern raised by this reviewer. First, the authors totally ignored my question about how they calculated SSB phase noise (of the references).

In response to the reviewer’s concern, the relevant data has been included in *Supplementary Information* as:

(on page S-5, line 2)

“4. SSB phase noise comparison

In Fig. 5b of the main text, the SSB phase noise values taken from references of 23, 31, 36, 37, and 38 were calibrated for the sake of fair comparison in two steps; First, each referenced SSB phase noise data set was quantified at the five distinct offsets in sequence at 1 Hz, 10 Hz, ..., and 10 kHz. Then, the data set was scaled with respect to a nominal carrier frequency of 0.1 THz by incorporating the scale factor of $-20 \log (f_c/0.1\text{THz})$ with f_c being the carrier frequency of each data set. The resulting raw numerical data adopted for comparison are shown in Table S2.”

Table S2. SSB phase noise data for Fig.5b. Data shows scaled data (raw data). Unit: dBc/Hz

Reference	Carrier frequency	Offset frequency				
		1 Hz	10 Hz	100 Hz	1 kHz	10 kHz
(i) This work	0.1 THz	-71	-73	-73	-73	-74
(ii) Freeman, J. R. et al. (ref. 23)	2.0 THz	N/A	N/A	-96 (-70)	-98 (-72)	-98 (-72)
(iii) Quraishi, Q. et al. (ref. 31)	0.3 THz	N/A	-36.5 (-27)	-49.5 (-40)	-56.5 (-47)	-63.5 (-54)
(iv) Zhang, S. et al. (ref. 36)	0.331 THz	-40.4 (-30)	-49.4 (-39)	-47.9 (-37.5)	-59.4 (-49)	-83.4 (-73)
(v) Tetsumoto, T. et al. (ref. 37)	0.3 THz	N/A	N/A	-44.5 (-35)	-69.5 (-60)	-94.5 (-85)
(vi) Tetsumoto, T. et al. (ref. 38)	0.3 THz	N/A	N/A	-39.5 (-30)	-67.5 (-58)	-119.5 (-110)

2-2) Second, the approach they described for fractional frequency instability does not lead to the results they are showing in Figure 5. This is a serious concern since Figure 5 benchmarks the authors' result relative to the state of the art and the listed parameters are not from the cited references and not reproducible based on the authors explanation. For each of the data points included in Figure 5 (both parts a and b), the authors should justify the values by their calculations.

The required justification has been included in the revised *Supplementary Information*;

(on page S-4, line 8)

“3. Fractional frequency instability comparison

The fractional frequency instabilities marked with asterisks in Fig. 5a of the main text were calculated by taking the ratio of the given spectral linewidth to the given carrier frequency. For more details, the measurement parameters taken in our calculation are summarized below in Table S1.”

The figure caption of Fig. 5 has been carefully revised as;

(on page 24, line 2)

“Figure 5 Performance comparison with state-of-the-art terahertz sources. a Short-term frequency instability at 1 s vs. operating frequency range; (i) this work, (ii-n) microcomb-based photomixing ($n=1-3$ refer to ref. 36–ref. 38), (iii-n) heterodyne mixing of cw lasers ($n=1-5$ refer to ref. 31–ref. 35), and (iv-n) QCLs ($n=1-3$ refer to ref. 21–23). The fractional frequency instabilities marked with an asterisk are extracted from linewidth data (see Methods). **b**, Approximate single-sideband (SSB) phase noise; (i) this work, (ii) QCL injection-locked to a frequency comb (referenced to a Rb clock)²³, (iii) heterodyne mixing of two CW lasers phase-locked to a frequency comb (referenced to a quartz crystal oscillator)³¹, (iv) 0.33 THz soliton microcomb stabilized to a hydrogen maser clock³⁶, (v) 0.30 THz soliton microcomb stabilized to a 10 GHz dielectric resonant oscillator (DRO)³⁷, (vi) 0.30 THz soliton microcomb stabilized to two optical Brillouin Stokes waves³⁸, and (vii) 0.1 THz signal of frequency multiplier chain (FMC) with up-conversion of an RF oscillator. For comparison, all carrier frequencies were scaled to 0.1 THz”

2-3) Detailed calculations should be given in a supplementary file with all the parameters taken from these references included. In providing this information the measurement parameters such as RBW, integration time, measurement time, the exact definition of linewidth, etc which directly impact the results should be clear and taken into account to provide a fair comparison. If there are two approaches taken for calculating fractional frequency instability, the authors should apply both approaches to their own results and prove with numbers that they are equal for their own data.

The required calculation results have been included in *Supplementary Information* together with all the parameters impacting the linewidth measurement;

(on page 14, line 17)

“Detailed information such as the linewidths, resolution bandwidths, and measurement time are given in Table S1 in Supplementary Information.”

(on page S-5, line 10)

“For more details, the measurement parameters taken in our calculation are summarized as below in Table S1.”

Table S1. Calculation of fractional frequency instability

Ref. Number	Emitter type	Locking method	Frequency reference	Carrier frequency	Linewidth (definition)	RBW	Meas. time	Estimated fractional frequency instability
Barbieri, S. et al. (ref. 21)	QCL	PLL to comb	Free-running comb	2.7 THz	1 kHz (N/A)	N/A	N/A	3.7×10^{-10}
Ravaro, M. et al. (ref. 22)	QCL	PLL to comb	Free-running comb	2.5 THz	1 kHz (N/A)	N/A	N/A	4.0×10^{-10}
Freeman, J. R. et al. (ref. 23)	QCL	IL to terahertz waves	Microwave reference	2.0 THz	100 Hz (N/A)	N/A	N/A	5.0×10^{-11}
Quraishi, Q. et al. (ref. 31)	Photomixer	2 CW lasers/PLL to comb	Quartz crystal oscillator	0.3 THz	2 Hz (FWHM)	1 Hz	1 s	6.7×10^{-12}
Steed, R. J. et al. (ref. 33)	Photomixer	2 CW lasers/PLL to comb	Microwave reference	0.3 THz	1 kHz (FWHM)	N/A	N/A	3.3×10^{-9}
Criado, Á. R. et al. (ref. 35)	Photomixer	2 CW lasers/IL to comb	Microwave reference	0.14 THz	10 Hz (FWHM)	10 Hz	N/A	8.3×10^{-11}

3) This response is unacceptable. The authors report a 2 mHz frequency resolution and synthesized linewidth as one of the main specs of their synthesizer. They cannot just mention the measurement method and state the final calculated value. They should include all the measurements and calculations done at 0.1, 0.6, and 1 THz that led to this 2 mHz linewidth calculation. They can put all the details in a supplementary file. They should compare the same measurements and calculations before and after stabilization.

This reviewer's suggestion has been reflected in the main text and *Supplementary Information* along with Fig. S2 & S3. Before and after thermal noise stabilization, the linewidth of the generated terahertz waves was measured 1.46 Hz and 2 mHz, respectively. This result confirms that the active suppression of the thermal fibre noise is essential to realize the superior stability and narrow linewidth to mHz level as demonstrated in our work. We have also supplemented the linear amplitude spectra of 0.10, 0.66, and 1.06 THz waves to clarify the 2 mHz FWHM linewidth calculations in Fig. S3.

(on page 7, line 9)

“... without the removal of the thermal noise of the fibre delivery line explained earlier (Fig. 2b), the spectral linewidth tends to broaden to 1.46 Hz, almost two orders of magnitude larger (Supplementary Information).”

(on page S-2, line 10)

“2. Spectral linewidth comparison.

Figure S2 presents the spectral linewidths of the terahertz waves measured with different stabilization conditions: (a) two free-running DFB lasers with no referencing to the source comb, (b) two comb lines before compensation of the fibre thermal noise, and (c) the same two comb lines after the thermal noise compensation. First, in the case of (a), the corresponding power spectrum shows no peak due to a large amount of thermal drift in the time domain as the DFB lasers are in a free-running state. Next, the case of (b) shows a coherent power spectrum having a spectral linewidth of 1.46 Hz (FWHM) with a Lorentzian fit. Finally, in case (c), with the suppression of the fibre thermal noise as described in the main text, the spectral linewidth shrinks to 2 mHz, which in fact reaches the resolution limit imposed by the Fourier-transform made in this measurement with a sample period of 500 s. Such drastic linewidth improvement is found to hold true for all the terahertz frequencies of 0.10, 0.66, and 1.06 THz tested in this investigation as presented in Figure S3.”

Figure S2. Comparison of the power spectra and spectral linewidths of the terahertz waves produced with three different conditions. a, Photomixing of two free-running distributed-feedback (DFB) lasers without comb-rooted injection locking. b, Photomixing of two comb lines before the suppression of the fibre thermal noise. c, Photomixing of the same two comb lines with thermal noise suppression. Note that the power spectrum of each case given on the right-hand side, in terms of the dB- and linear scale, is the result of the fast Fourier transform of a data set of 500-s sampling.

Figure S3 Spectral linewidths of terahertz waves. The terahertz frequencies of 0.10, 0.66, and 1.06 THz are determined with a resolution and a linewidth of 2 mHz, which is limited by the FFT limit.

5) The statement “The current noise floor of our synthesizer, being limited by the emitted power of terahertz waves and the thermal noise of the PCA, will be further improved in the near future with the advent of high-power photomixers and low-noise PCAs.” is misleading. The authors are using commercially available photomixers and PCAs like what many other references have used. Any advances in the radiation power and noise performance of commercially available photomixers and PCAs would have the same impact on all synthesizers demonstrated by the authors and other groups since all of them are based on the same frequency conversion process. This statement suggests that only the authors’ synthesizer would benefit from higher performance photomixers and PCAs, which is not true.

We appreciate the reviewer’s comment, but we need to justify our claim further as;

Fig. SS3 Phase noise evaluation of our terahertz synthesizer

The phase noise of terahertz waves generated via a photomixer and detected by a PCA is affected by two causes: (1) laser phase noise and (2) detector noise. In our synthesizer, the phase noise becomes limited dominantly by the detector noise - Johnson noise at the photoconductive antenna - over 1 Hz offset frequency (red curve, Fig. SS3 above). As a result, **the noise coming from the comb lines (blue curve) causes no significant effects on the overall phase noise of the terahertz waves as discussed in Methods as;**

“Noise analysis in PCA heterodyne detection. The phase noise floor of the real terahertz signal (red curve, Fig. 2c) is contributed mainly by three distinct noise sources, each being distinguished as N1, N2 and N3. Note N1 represents the TIA’s intrinsic noise, while N2 and N3 refer to the PCA’s photocurrent shot noise and thermal (Johnson-Nyquist) noise, respectively. The 100 kHz beatnote signal delivered from the PCA carries an

average photocurrent of 3.6 nA, producing a shot noise (N_2) of $34 \text{ fA}/\sqrt{\text{Hz}}$ with a phase noise amplitude of $-100 \text{ dBc}/\text{Hz}$. Meanwhile, the thermal noise (N_3) of the PCA and the intrinsic TIA noise (N_1) are measured using an FFT analyzer (SR760, SRS). With the TIA's input placed in an open-circuit state with a transimpedance gain of $2 \times 10^7 \text{ V}/\text{A}$, the TIA's intrinsic noise (N_1) is found to be $35 \text{ fA}/\sqrt{\text{Hz}}$ at a 100 kHz offset, which causes a phase noise floor of $-100 \text{ dBc}/\text{Hz}$. Next, with the PCA output being connected to the TIA input, the thermal noise (N_3) of the PCA is monitored to be $560 \text{ fA}/\sqrt{\text{Hz}}$ which is predicted to be the most dominating noise floor of $-76 \text{ dBc}/\text{Hz}$. The thermal noise (N_3) shows no dependence on the optical pump power induced on the PCA in the 0 – 30 mW range, which is known as a typical character of the thermal noise created from the PCA54. The estimated thermal noise level of $-76 \text{ dBc}/\text{Hz}$ agrees well with the actually measured result of $-73 \text{ dBc}/\text{Hz}$ with a minimal unaccounted 3 dB discrepancy. Note that the PCA noise limit may also be mitigated by either increasing the repetition rate of the source comb or employing a sensitive Schottky diode detector instead of the PCA, which will be attempted in our near-future follow-up investigations.”

Exemplary works for microcomb-based terahertz generation

On the other hand, for the reference papers shown above, the phase noise level is affected mainly by the laser phase noise, of which method is similar to our work in that comb lines are used to coherently connect time/frequency standards to the terahertz region. Among the RF frequency references (the H-maser clock, a DRO clock, and an ultra-stable Brillouin light) used in the recent works to stabilize the generated terahertz waves; the H-maser-referenced terahertz wave shows the best frequency stability of 2×10^{-13} at 1-s averaging. This indicates that the phase noises of generated terahertz waves are restricted mainly by the laser phase noises originating from the time/frequency references. In other words, the power increase of photocurrent has no effects on the phase noise improvement, lowering the noise floor only at a higher frequency offset (over 1-MHz offset frequency).

In this regard, our claim “the current noise floor of our synthesizer, being limited by the emitted power of terahertz waves and the thermal noise of the PCA, will be further improved in the near future with the advent of high-power photomixers and low-noise PCAs.” is appropriate as our outlook to exploit the best stability of the optical cavity fully.

6) The parameters that authors have used to benchmark their results (those shown in Figure 5) are not used by most of the references they have listed (as authors also agree and mention in their response to my second comment). Since the authors calculate the fractional frequency stability and SSB phase noise for other references, they can also calculate frequency noise power spectral density (FNPSD). In addition, it is very important to calculate the frequency noise power spectral density (FNPSD) of the demonstrated synthesizer as it is a better benchmark for stability of the synthesizer

As requested, the FNPSD calculation of our synthesizer has been included in the revised *Supplementary Information*.

(on page S-2, line 2)

“The phase noise spectra given in Fig. 2c of the main text are converted to their corresponding frequency noise power spectral densities (FNPSDs) as presented in Figure S1. The FNPSD plots provide additional information for the performance comparison between our terahertz synthesizer and other state-of-the-art counterparts.”

Figure S1 Frequency noise power spectral density (FNPSD) plots.

7) The authors say “overall performance, particularly in terms of the maximum tuning range and frequency noise, is limited mainly by the signal-to-noise ratio of the used photomixer and PCA” however, both photomixers and PCAs today have much larger tuning range than 1 THz (more than 5 THz). The authors should discuss the physical limitations of the optical combs generated by their presented scheme in providing higher terahertz frequencies.

We admit that there was a misunderstanding about the relationship between the tuning range and SNR. Thus, for clarity, we have split the discussion on the maximum tuning range and frequency noise, in the revised manuscript as:

(on page 9, line 12)

“The maximum tuning range is currently set at 4 THz in consideration of the spectral bandwidth of the source comb, which can readily be extended to more than 10 THz by supercontinuum generation using a highly nonlinear fiber⁵².”

(on page 9, line 14)

“Meanwhile, the frequency noise is limited mainly by the signal-to-noise ratio of the used photomixer and PCA, which will be alleviated gradually in the near future with the advent of more high-power photomixers and high-sensitivity emitters.

(on page 7, line 17)

“Higher generated terahertz frequencies suffer worse stabilities, being attributable to the phase noise contributed by not only the photomixer but also the PCA used for terahertz detection (Methods).”

8) ‘15-digit short-term stability’ is very vague in describing the actual physical performance. The authors should mention the exact numbers in terms of both accuracy and time: for example, 4.4×10^{-15} at 1 s integration.

The reviewer’s suggestion has been reflected throughout the revised manuscript as;

(on page 1, line 18) “.... 4.4×10^{-15} at 1-s integration.”

(on page 3, line 10) “.... 4.4×10^{-15} at 1-s integration ...”

(on page 9, line 8) “... The frequency instability turns out to be 4.4×10^{-15} at 1-s integration, ...”

Further, the title of the revised manuscript has been changed to “Photonic comb-rooted synthesis of ultra-stable terahertz frequencies” in consideration of the reviewer’s suggestion not to use the term of “15-digit short-term stability” due to its vagueness.

REVIEWERS' COMMENTS

Reviewer #3 (Remarks to the Author):

The authors' revisions and responses show a significant improvement in the manuscript and better justify the reported results. I have no reservation to accept the manuscript.

[Authors' responses to Reviewer #3]

The authors' revisions and responses show a significant improvement in the manuscript and better justify the reported results. I have no reservation to accept the manuscript.

The authors very much appreciate the reviewer's positive evaluation.